# Position: Solve Layerwise Linear Models First to Understand Neural Dynamical Phenomena (Neural Collapse, Emergence, Lazy/Rich Regime, and Grokking)

**Yoonsoo Nam** [* 1]  **Seok Hyeong Lee** [* 2]  **Clementine Carla Juliette Domine** [3]  **Yeachan Park** [4]  **Charles London** [5]
**Wonyl Choi** [6]  **Niclas Alexander Goring** [1]  **Seungjai Lee** [7]

## Abstract

In physics, complex systems are often simplified into minimal, solvable models that retain only the core principles. In machine learning, layerwise linear models (e.g., linear neural networks) act as simplified representations of neural network dynamics. These models follow the dynamical feedback principle, which describes how layers mutually govern and amplify each other's evolution. This principle extends beyond the simplified models, successfully explaining a wide range of dynamical phenomena in deep neural networks, including neural collapse, emergence, lazy and rich regimes, and grokking. In this position paper, we call for the use of layerwise linear models retaining the core principles of neural dynamical phenomena to accelerate the science of deep learning.

## 1. Introduction

Physicists often build intuition about complex natural phenomena by abstracting away intricate details—for instance, modeling a cow as a sphere (Kaiser, 2014), linearizing pendulum motion (Huygens, 1966), or using simplified frameworks like Hopfield networks to explore associative memory and network dynamics (Hopfield, 2007). These abstractions enable tractable analysis while preserving the core principles of the system. A good model not only allows analytical solutions but also captures the underlying principle, often providing insights and extending their utility beyond their formal limits.

Deep neural networks (DNNs) are complex dynamical systems, with non-linear activations (e.g., ReLU) posing major challenges in analysis. Without non-linear activations, DNNs become layerwise linear models (e.g., linear neural networks), often underappreciated as simple product of matrices for their lack of expressivity. However, **the dynamics of layerwise linear models are non-linear** (Saxe et al., 2014), allowing them to explain key DNN phenomena such as emergence (Nam et al., 2024) observed in large language models (Brown et al., 2020), neural collapse (Mixon et al., 2020) observed in image classification tasks (Papyan et al., 2020), lazy/rich regimes (Dominé et al., 2025) observed in extreme limits (Jacot et al., 2018; Chizat et al., 2019), and grokking (Kunin et al., 2024) observed in algorithmic tasks (Power et al., 2022), and more as **dynamical phenomena of layerwise structure**.

Contrary to most non-linear models, the dynamics of layerwise linear models, under suitable assumptions, are exactly solvable (e.g., Baldi & Hornik (1989); Saxe et al. (2014); and Braun et al. (2022)). These solutions allow mathematical interpretations that extend beyond the limits of the models. In this paper, we propose the dynamical feedback principle (Section 3) as a unified principle for understanding layerwise linear models, and review how they accurately represent DNN dynamics to argue our position:

**Position: We call for new studies to prioritize the dynamics of layerwise structure as the key feature over the non-linear activations, until its limits are fully explored.**

Layerwise linear models help identify whether the dynamics of layerwise structure drives a phenomenon, demystifying perceptions of the complex roles of datasets, logic, and non-linear activations. Their solvability enables a theoretical approach, while their simplicity makes them accessible to a broader audience, accelerating the science of deep learning.

---

[*]Equal contribution [1]Department of Theoretical Physics, University of Oxford, Oxfordshire, United Kingdom [2]Center for Quantum Structures in Modules and Spaces, Seoul National University, Seoul, South Korea [3]Gatsby Computational Neuroscience Unit, University College London, London, United Kingdom [4]Department of Mathematics and Statistics, Sejong University, Seoul, South Korea [5]Department of Computer Science, University of Oxford, Oxfordshire, United Kingdom [6]Department of Computer Science, Boston University, Massachusetts, United States of America [7]Department of Mathematics, Incheon National University, Incheon, South Korea. Correspondence to: Seungjai Lee <seungjai.lee@inu.ac.kr>.

*Proceedings of the 42nd International Conference on Machine Learning*, Vancouver, Canada. PMLR 267, 2025. Copyright 2025 by the author(s).

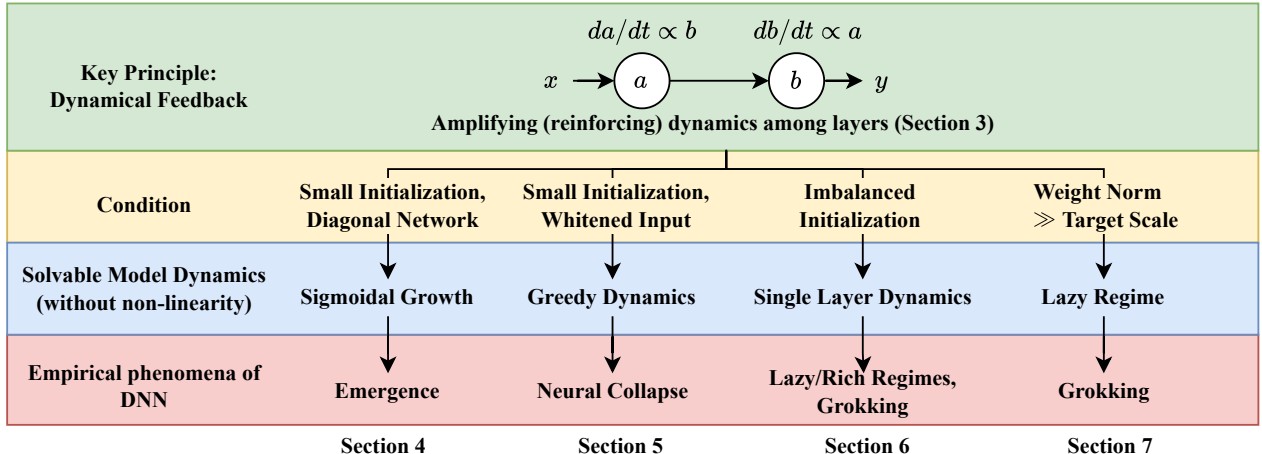

Figure 1. **Paper outline** For a systematic presentation of various works on layerwise linear models, we begin the section by building intuition on how the key principle (green) behaves under a condition (yellow). We then formalize this intuition as a key property (blue) using a solvable layerwise linear model. Finally, we discuss how this property from the layerwise linear model extends to describe an empirical phenomenon in DNNs (red).

## 1.1. Contributions

- We propose dynamical feedback principle (Section 3) as a novel unifying principle for understanding seemingly uncorrelated DNN phenomena (neural collapse, emergence, lazy/rich regimes, and grokking) as consequences of dynamics under layerwise structure.

- We review solvable layerwise linear models (Sections 4 to 7) and discuss their significance through the principle and intuitive derivations.

- We solve a toy model that offers intuitive insight on various lazy regimes and methods to induce rich regimes (Section 7).

## 1.2. Paper outline

Figure 1 illustrates the paper outline. In Section 3, we introduce the dynamical feedback principle, which unifies the explanation of seemingly unrelated DNN phenomena.

In Sections 4 to 7, we first demonstrate how the principle applies under specific conditions, then formalize the intuition using a solvable layerwise linear model, **prioritizing intuitive derivations over rigor** found in the references. Finally, we explore how the principle extends beyond layerwise linear models to explain phenomena in practical DNNs.

In the discussion, we argue that solvable layerwise linear models effectively capture the dynamics of DNN through the feedback principle, making them essential for advancing DNN theory, hence our position. We conclude with promising research directions. For a **glossary** of notations and terms (emphasized in *italic*), see Appendix A.

## 2. Setup

Throughout the paper, unless explicitly stated, we assume the input features $x \in \mathbb{R}^d$ to be $d$ dimensional independent zero-mean Gaussian random variables. We only consider gradient flow (GF) and assume that the training dataset follows a certain probability distribution for analytic simplicity.

**Target function** We assume a realizable target function $f^*$ with *target scales* $S_i \in \mathbb{R}$ for $i = 1, \ldots, d$:

$$f^*(x) = \sum_{i=1}^{d} x_i S_i. \tag{1}$$

**Loss** We consider the mean square error (MSE) loss

$$\mathcal{L} := \frac{1}{2} \mathbf{E} \left[ (f^*(x) - f(x))^2 \right]. \tag{2}$$

**Models** We define *layerwise linear models* as 2-layer models whose outputs are multilinear with respect to any layers of parameters. Examples include the variants of linear neural networks, illustrated in Figure 2:

$$f_{a,b}^{(diag)}(x) = \sum_{i=1}^{d} x_i a_i b_i, \quad a, b \in \mathbb{R}^d, \tag{3}$$

$$f_{W_1,W_2}^{(lnn)}(x) = x^T W_1 W_2, \quad W_1 \in \mathbb{R}^{d \times p}, W_2 \in \mathbb{R}^{p \times c}. \tag{4}$$

A linear model with no hidden layer is used as a base model for comparison:

$$f_{\theta}^{(lin)}(x) = x^T \theta, \quad \theta \in \mathbb{R}^d. \tag{5}$$

Like linear models, layerwise linear models do not require linearity with the input features $x_i$, though we use them for demonstrative purposes. See Appendix B for details.

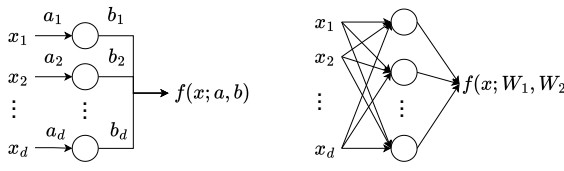

(a) Diagonal linear neural network      (b) Linear neural network

*Figure 2.* **Layerwise linear models.** Layerwise linear models (Appendix B) include, but are not limited to, **(a):** diagonal linear neural networks (Equation (3)) and **(b):** linear neural networks (Equation (4)). The layerwise structure leads to distinct dynamics compared to linear models (Equation (5)).

## 3. The Dynamical Feedback Principle

Dynamical feedback principle describes how the magnitude of one layer mutually scales the rate of change in other layers (Figure 1). The dynamical feedback principle arises from the gradient descent dynamics under layerwise structure. For example, the GF equation of diagonal linear neural network is

$$\frac{da_i}{dt} = -b_i \mathbf{E}[x_i^2](a_i b_i - S_i), \quad \frac{db_i}{dt} = -a_i \mathbf{E}[x_i^2](a_i b_i - S_i).$$
(6)

The product of parameters (i.e., layerwise structure) creates an *amplifying dynamics*: the size of $a_i$ governs the rate of change for $b_i$, and vice versa, yielding a **dynamical feedback**. To highlight its significance, we compare the gradient equation to that of the linear model:

$$\frac{d\theta_i}{dt} = -\mathbf{E}[x_i^2](\theta_i - S_i).$$
(7)

In linear models, parameters evolve depending only on their distances to respective target scales ($\theta_i - S_i$), lacking the feedback with other parameters. The following sections focus on exploring the consequences of feedback.

**Conservation of magnitude difference** Because of the commutativity of products in Equation (3), we identify a conserved quantity in Equation (6) as

$$a_i \frac{d}{dt} a_i - b_i \frac{d}{dt} b_i = 0 \quad \Rightarrow \quad a_i^2 - b_i^2 = \mathcal{C}_i,$$
(8)

where $\mathcal{C}_i$ is determined at initialization. We can extend the argument for linear neural networks to show that $W_2 W_2^T - W_1^T W_1$ is conserved (Appendix C).

## 4. From Amplifying Dynamics to Emergence

In Equation (6), when $a_i \approx b_i$, the dynamical feedback exhibits rich-get-richer (poor-get-poorer) property: a larger (smaller) $|a_i|$ results in a faster (slower) evolution of $b_i$, which amplifies (attenuates) in positive (negative) feedback. In this section, we formalize this property as *sigmoidal growth* (delayed saturation) of $a_i b_i$ in diagonal linear neural networks, which explains the *emergence* – or sudden change in task or subtask performance – observed in large language models (Ganguli et al., 2022; Srivastava et al., 2023; Wei et al., 2022; Chen et al., 2023).

### 4.1. Sigmoidal dynamics and stage-like training

We consider dynamics of diagonal linear neural networks (Equation (6)) with small and equal initialization; $a_i(0) = b_i(0)$ and $0 < a_i(0) \ll 1$. The dynamics decouples into $d$ independent *modes*, where each mode follows a sigmoidal saturation:

$$a_i(t)b_i(t)/S_i = \frac{1}{1 + \left(\frac{S_i}{a_i(0)b_i(0)} - 1\right)e^{-2S_i \mathbf{E}[x_i^2]t}}.$$
(9)

This contrasts to the exponential saturation of modes in the linear model with $\theta_i(0) = 0$ (Equation (7)):

$$\theta_i(t)/S_i = 1 - e^{-\mathbf{E}[x_i^2]t}.$$
(10)

See Appendix C for the derivation of both models. For a general derivation when $a_i \neq b_i$, see Appendix A of Saxe et al. (2014).

Figure 3 demonstrates the difference between the dynamics of two models. While a mode in linear models saturates immediately after initialization (Figure 3(a)), that of diagonal linear neural network saturates **after a delay**. The small value of $a_i b_i$ delays the saturation as small $a_i$ downscales the gradient of $b_i$ and vice versa (feedback principle). After sufficient evolution, the feedback principle abruptly saturates $a_i b_i$ (Figure 3(b)).

**Stage-like training** In Figure 3(b), the first mode (blue) saturates fully while the next mode remains negligible, contrasting with the concurrent saturation of modes in linear models (Figure 3(a)). The sequential learning of modes or the stage-like training occurs in layerwise models when saturation time scales ($1/(S_i \mathbf{E}[x_i^2])$) differ sufficiently; the sigmoidal dynamics (Equation (9)) delays slower modes long enough for faster modes to fully saturate.

### 4.2. Application 1: Saddle-to-saddle learning

Saddle-to-saddle learning refers to sudden drops and plateaus of the loss during training (Jacot et al., 2022; Pesme & Flammarion, 2023; Atanasov et al., 2025). We can describe the DNN phenomenon with the stage-like training

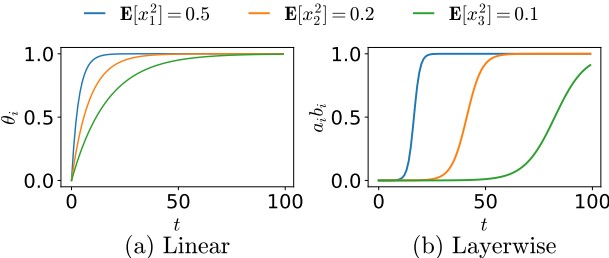

(a) Linear         (b) Layerwise

*Figure 3.* **Dynamics of the linear model and the diagonal linear neural network.** The colored lines show the saturation curves of modes with different variances for **(a):** the linear model (Equation (10)) and **(b):** the diagonal linear neural network (Equation (9)) with $S_i = 1$. For the linear model, all $\theta_i$'s saturate from $t = 0$ only differing in the saturation speed. For the layerwise model, $a_i b_i$'s show delayed saturations depending on $\mathbf{E}[x_i^2]$, learning the modes in sequences.

of diagonal linear neural network with varying $S_i \mathbf{E}[x_i^2]$. A drop maps to the abrupt saturation of a feature, while a plateau to the delay before the saturation of the next feature.

### 4.3. Application 2: Emergence

Emergence is an empirical observation that larger languages models suddenly gain performance when using more data or parameters (Wei et al., 2022).[1] Emergence has been considered challenging to explain through *"analysis of gradient-based training"* (Arora & Goyal, 2023), and studies have been focused on defining basic *skills* and how they cumulate to a sudden improvement in more complex abilities (Arora & Goyal, 2023; Chen et al., 2023; Yu et al., 2024; Okawa et al., 2024).

**Diagonal linear neural network with prebuilt skills** In Nam et al. (2024), the authors modeled emergence as a **dynamical outcome** of sigmoidal growth. They studied the multitask parity problem, consisting of parity tasks (skills) whose frequencies follow a power-law distribution. They used diagonal linear neural network with the input features ($x_i$ in Equation (3)) replaced by the skill functions $g_k(x)$:

$$f(x) = \sum_{k=1}^{p} a_k b_k g_k(x), \qquad f^*(x) = S \sum_{k=1}^{p^*} g_k(x). \quad (11)$$

The skill functions $g_k(x)$ are prebuilt features that map the inputs to a scalar by calculating the spare parity function. The constants $p$ and $p^*$ are the number of skill functions in the model and target function, respectively. In their setup, the model learns the skills through sigmoidal dynamics (Equation (9)), prioritizing the more frequent skills (with larger $\mathbf{E}[g_k^2(x)]$) first — thus the time emergence.

---

[1]Emergence differs from grokking in that it does not account for the **gap** between achieving prefect training and test accuracies.

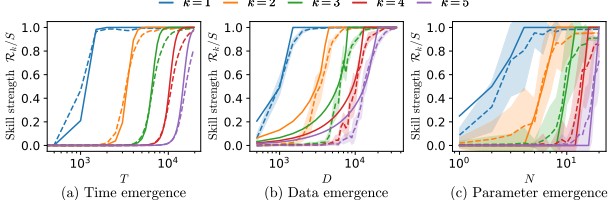

(a) Time emergence    (b) Data emergence    (c) Parameter emergence

*Figure 4.* **Predicting emergence with layerwise linear model (Figure 1 from (Nam et al., 2024)).** The skill strength $\mathcal{R}_k = \mathbf{E}[f(x)g_k(x)]/\mathbf{E}[(g_k(x))^2]$ measures the linear correlation between the $k^{\text{th}}$ skill function and the learned function or how well the skill is learned. Each color represents a different skill. The solid lines show the theoretical layerwise model (Equation (11)) calibrated on the first skill, while the dashed lines represent the empirical results of a 2-layer neural network trained on the multitask sparse parity problem.

The decoupled skills in the model induce decoupled dynamics: the $k^{\text{th}}$ skill is learned only if $g_k(x)$ exists in the model ($k \leq p$) and a corresponding sample (from the $k^{\text{th}}$ skill) appears in the training set. This leads to abrupt shifts (emergence) in learning — for instance, the $k^{\text{th}}$ function is suddenly acquired as $p$ increases from $k - 1$ to $k$. Similar discontinuities arise in the data, depending on the likelihood of encountering a sample from the $k^{\text{th}}$ skill. In Figure 4, the model (Equation (11)) accurately captures the emergent dynamics in a 2-layer neural network (with ReLU activation) for time, data, and the number of parameters. See Appendix D or Nam et al. (2024) for details.

### 4.4. Application 3: Scaling laws

Scaling laws refer to how the performance of large language models scales as a power law with additional resources (e.g., compute, data, parameters) (Hestness et al., 2017; Kaplan et al., 2020). Michaud et al. (2023) proposed that scaling laws arise from the emergent learning of numerous subtasks or skills whose frequencies follow a power law. Consequently, Nam et al. (2024) predicted the scaling laws observed in 2-layer neural networks trained on multitask sparse parity problem with Equation (11).

**Intuition** The authors of Nam et al.(2024) derived emergence and scaling laws (for time, data, and parameters), relying heavily on the a priori given skill functions $g_k$ (Equation (11)) and their decoupled dynamics. But why does a neural network — without any information on the skill functions or decoupled dynamics — show similar behavior? This behavior can be attributed to the layerwise structure and the power-law distribution of skill frequencies, which together induce **stage-like training** in neural networks, where skills are feature-learned on different time scales. This effectively decouples feature learning, allowing the network to distinguish features and justifying the use of prebuilt skill

functions in Equation (11). See Appendix D or Nam et al. (2024) for a detailed discussion.

# 5. From Greedy (Low-Rank) Dynamics Toward Salient Features to Neural Collapse

In Section 4, we demonstrated that sigmoidal saturation (Equation (9)) in diagonal linear neural networks prioritizes learning features that are more correlated (with larger $S_i\mathbf{E}[x_i^2] = \mathbf{E}[x_i y_i]$). In this section, by building on Saxe et al. (2014), we show how the preference toward correlated features leads to a *low-rank bias* (**greedy dynamics**) in linear neural networks. We then draw on Mixon et al. (2020) to explain *neural collapse* — the collapse of last-layer features into a low-rank structure —observed in vision classification tasks (Papyan et al., 2020).

## 5.1. Greedy dynamics

The dynamics of linear neural networks is challenging to solve in general. Assuming small initialization and whitened input ($\mathbf{E}[xx^T] = I$), Saxe et al. (2014) and following works (Ji & Telgarsky, 2019; Arora et al., 2019a; Lampinen & Ganguli, 2019; Gidel et al., 2019; Tarmoun et al., 2021) solved the exact dynamics of linear neural networks. The dynamics **decouples** into $c$ independent modes similar to that of diagonal linear neural network (Appendix C):

$$\frac{\alpha_i^2(t)}{\rho_i} = \frac{1}{1 + \left(\frac{\rho_i}{\alpha_i^2(0)} - 1\right)e^{-2\rho_i t}}, \qquad (12)$$

where $\alpha_i := u_i^T W_1 W_2 v_i$, and $u_i, \rho_i, v_i$ are the $i^{\text{th}}$ left singular vector, singular value, and the right singular vector of the correlation matrix $\mathbf{E}\left[xf^*(x)^T\right] = UPV$, respectively.

Note the similarity between Equation (9) and Equation (12), where $a_i b_i$ and $S_i$ in Equation (9) are replaced by $\alpha_i^2$ and $\rho_i$ in Equation (12), respectively. Linear neural networks automatically **identify** the linear combination of input features **most correlated** (largest $\rho_i$) with the target (salient features). As discussed in stage-like training, modes are learned sequentially based on the order of $\rho_i$ or their saliency. This bias toward salient features introduces a preference for minimum rank, as only target-relevant $c$ (the dimension of the output) modes are trained, even though $W_1$ can reach a rank of $\min(d, p)$. We will refer to the tendency toward learning the most target-relevant features first as the **greedy dynamics**.

## 5.2. Application: Neural collapse

Neural collapse (NC) (Papyan et al., 2020) describes the phenomenon where last-layer features form a simplex equiangular tight frame (ETF) – orthogonal vectors projected at the complement of the global mean. For instance, last layer feature vectors of ResNet18 trained on CIFAR10 converge to 10 orthogonal vectors (9-simplex ETF), corresponding to the number of classes (Papyan et al., 2020). See Appendix E or (Papyan et al., 2020) for the formal definition.

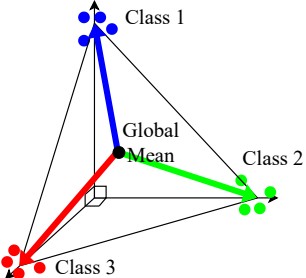

*Figure 5.* **Illustration of neural collapse.** In NC, the last layer feature vectors (the post-activation of the penultimate layer), illustrated as colored dots, cluster by their class mean vector, illustrated as colored arrow, and form a simplex ETF structure (orthogonal vectors projected at the compliment of the global mean vector).

The NC gained attention as the last layer feature vectors occupied only $c$ dimensions out of $p$ dimensional feature space, where $c$ is the number of classes and $p$ is the width of the last layer. Mixon et al. 2020 and similar works (Fang et al., 2021; Han et al., 2022) studied the dynamics of NC through the unconstrained feature model (UFM). The UFM trains a product of two matrices where one matrix represents the features and the other matrix represents the last layer weights, a special case of linear neural networks. See also relevant works on matrix factorization (Arora et al., 2019b; Li et al., 2021).

**Informal derivation of NC** We show a non-rigorous derivation of NC and discuss why feature vectors of linear neural network $XW_1$, where $X \in \mathbb{R}^{n \times d}$ is the input feature matrix for $n$ training datapoins, collapse into a rank $c$ matrix. For rigorous theorems, see Mixon et al. (2020) or Fang et al. (2021). First, we assume sufficiently small initialization such that the conserved quantity (Equation (8)) is $W_1(0)^T W_1(0) - W_2(0)W_2(0)^T \approx 0$. After training,

$$XW_1(\infty)W_2(\infty) = Y, \qquad (13)$$

where $Y \in \mathbb{R}^{n \times c}$ is the label matrix with one-hot vector as the labels, and $W_i(\infty)$ the trained matrices. Since $Y$ is a rank $c$ matrix, $W_2(\infty)$ is also rank $c$. The small initialization condition $W_1^T W_1 - W_2 W_2^T \approx 0$ ensures that $W_1$ and $W_2$ have the same rank, resulting in the feature matrix $XW_1$ having rank $c$ instead of $\min(n, p)$. The matrix $XW_1$ collapses into $c$ orthogonal directions, with each direction trivially mapping to a class to satisfy Equation (13).

**Intuition**   Loosely speaking, $W_1^T W_1 - W_2 W_2^T \approx 0$ is analogous to the small initialization in Saxe et al. (2014).[2] From the dynamics perspective, the greedy dynamics of layerwise models (Equation (12)) sends gradients toward only the relevant $c$ directions (modes), effectively limiting the rank of the feature matrix ($XW_1$).

Practical DNNs are often initialized with small weights and are expressive enough to fit the training set without using all layers. Due to their layerwise structure and expressiveness, the greedy dynamics toward minimal rank likely persist even with nonlinear activations, leading to neural collapse (Jacot et al., 2024).

**Greedy dynamics and depth**   In deeper networks, the additional parameter products intensify amplifying dynamics, sharpening sigmoidal saturation into a Heaviside-like step functions, and widening the gap between more and less correlated features. Under mild assumptions, such models admit analytic solutions (Saxe et al., 2014; Lyu & Zhu, 2023; Súkeník et al., 2023; Kunin et al., 2024), revealing biases toward $L_1$-norm solutions and stronger reliance on the network's initial representation. The analysis also offers an explanation for the Lottery Ticket Hypothesis (Frankle & Carbin, 2019) where most parameters can be pruned with minimal performance loss.

## 6. From Layer Imbalance to Lazy/Rich Regimes

So far, we assumed that layers have similar magnitudes ($a_i \approx b_i$), where the feedback principle creates positive or negative feedback on their change. However, for heavily imbalanced layers (e.g., $a_i \gg 1 \gg b_i$), the heavier layer receives negligible gradients, remaining nearly fixed and disrupting nonlinear reinforcement between layers. In this section, we formalize the role of layer imbalance in linear neural networks and review Kunin et al. (2024) to show how layer imbalance can control *lazy/rich regimes* (Chizat et al., 2019; Dominé et al., 2025) — a linear/non-linear dynamics — and *grokking* (Power et al., 2022) — a delayed generalization — in neural networks.

### 6.1. $\lambda$-balanced assumption

Deriving explicit dynamics for linear neural networks with general initialization is challenging. However, we can formalize the intuition of layer imbalance as a solvable model under the following condition (Fukumizu, 2000; Arora et al., 2018; Braun et al., 2022; Dominé et al., 2025; Kunin et al., 2024; Varre et al., 2024):

$$W_2 W_2^T - W_1^T W_1 = \lambda I, \tag{14}$$

---

[2]Large initialization can achieve $W_1^T W_1 - W_2 W_2^T \approx 0$ and similar dynamics (Braun et al., 2022; Dominé et al., 2025).

where $I$ is an identity matrix. Recent analytical models (Dominé et al., 2025; Kunin et al., 2024; Tu et al., 2024; Xu & Ziyin, 2024) solved the exact dynamics for a square ($d = c$) linear neural network satisfying Equation (14) and confirmed the intuition on layer imbalance. They showed that smaller $|\lambda|$ (balanced layers) leads to nonlinear dynamics similar to greedy dynamics even with large weight initialization, while larger $|\lambda|$ (imbalanced layers) results in linear dynamics where only the lighter layer is trained (Figure 6(a)). For details and their relationship to the architecture, see Appendix F.

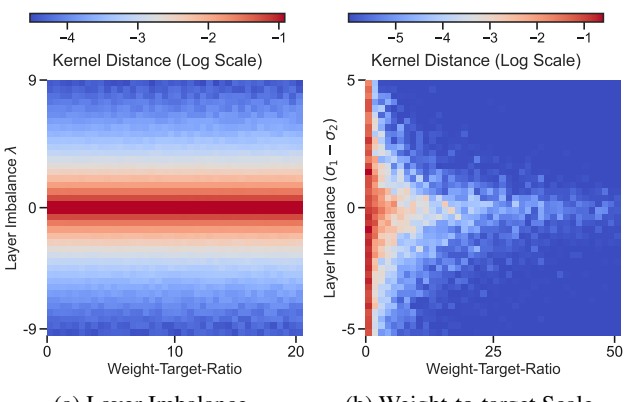

(a) Layer Imbalance          (b) Weight-to-target Scale

*Figure 6.* **Two ways to get lazy dynamics.** Figures show the changes in NTK (measure of non-linear dynamics) of linear neural network trained on randomized regression. **(a)**: In Section 6, when Equation (14) holds, layer imbalance governs the transition between rich (red) and lazy (blue) dynamics. **(b)**: In Section 7, when Equation (14) does not hold, larger weight-to-target ratio pushes the dynamics toward lazy regime, although layer imbalance still has an effect. For details of the experiment, see Appendix F.

### 6.2. Application 1: Target learning a layer with layer imbalance

*Lazy* regimes (Chizat et al., 2019; Woodworth et al., 2020; Azulay et al., 2021) refer to when neural networks shows a linear dynamics (Equation (7)) while *rich* regime refers to a sufficient deviation from it (e.g., non-linear dynamics of Equation (6)).

In 2-layer linear neural networks, learning only $W_1$ or $W_2$ through layer imbalance results in linear dynamics (lazy). In 2-layer neural networks, the non-linear activations (e.g., ReLU) **breaks the symmetry** between layers: updating only the last layer results in linear dynamics, whereas updating only the first layer can facilitate feature learning. Kunin et al. (2024) demonstrated that the upstream initialization — heavier later layers — promotes the learning in the earlier layers of convolutional neural networks and improves their interpretability and feature learning.

## 6.3. Application 2: Controlling grokking with layer imbalance

Grokking (Power et al., 2022) refers to a delayed generalization. Grokking was considered as an artifact of algorithmic dataset, but recent studies (Kumar et al., 2024; Lyu et al., 2024; Kunin et al., 2024) showed that grokking is a transition from a lazy overfitting regime to a rich generalizing regime. See Appendix G for details. Kunin et al. (2024) used upstream initialization to focus learning on earlier layers, reducing the generalization delay in a transformer trained on a modular arithmetic task. See Appendix F or Kunin et al. (2024) for details.

**Intuition** The dynamical feedback principle, valid even with non-linear activations, enables layer imbalance to control which layer is trained significantly. Thus, the layer imbalance can control lazy/rich regimes and grokking.

# 7. From weight-to-Target Ratio to Grokking

As discussed in Section 6, sufficient layer imbalance can mitigate feedback effects (Figure 6(a)). At the same time, Equation (6) suggests that mitigation can also arise when $a_i(0)b_i(0)$ closely matches $S_i$, allowing dynamics to complete before feedback takes effect (Figure 6(b)). In this section, we use linear neural networks to formalize this intuition through the *weight-to-target ratio* that can explain two lazy regime methods — NTK initialization (Jacot et al., 2020) and target downscaling (Chizat et al., 2019; Geiger et al., 2020) — within a single framework. We then empirically show that reducing the weight-to-target ratio eliminates the delay in grokking, accelerating generalization.

## 7.1. Toy model for NTK initialization and target downscaling

We assume a 2-layer $p$-wide linear neural network with scalar input $x$ and scalar output ($d = c = 1$):

$$f(x) = \frac{x}{Z} \sum_{i=1}^{p} a_i b_i, \quad f^*(x) = xS, \quad (15)$$

where $x \in \mathbb{R}$, $S > 0$, and $Z \in \mathbb{R}$ is a normalization or output rescaling constant. In Appendix H, we solve the exact dynamics of Equation (15) for any initialization (e.g., Gaussian distribution). For simplicity, we demonstrate a special case where $a_i(0)$ and $b_i(0)$ are initialized from $\{-1, 1\}$ with equal probabilities.

The dynamics can be decomposed into two modes: the **aligned** mode $\mathcal{I}_+ = \{i : a_i(0)b_i(0)S > 0\}$, and the **anti-aligned** mode $\mathcal{I}_- = \{i : a_i(0)b_i(0)S < 0\}$, where $i \in$

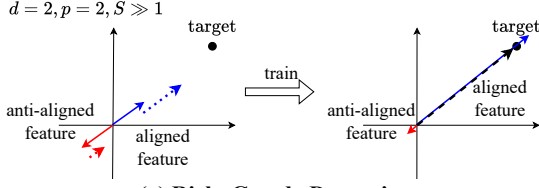

**(a) Rich: Greedy Dynamics**

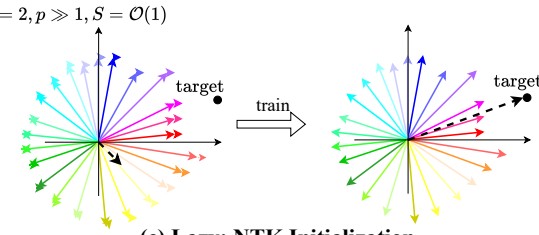

**(b) Lazy: Target Downscaling**

**(c) Lazy: NTK Initialization**

*Figure 7.* **Illustration of rich vs. lazy dynamics.** The colored arrows represent features (e.g., $a_i b_i x_i$ in Equation (15)), the dotted arrows represent their gradients, and the black dashed arrow represents the learned function in function space. **(a):** With smaller initialization and larger $S$, amplifying dynamics reinforce the growth of the aligned feature (larger parameters lead to faster growth) while mitigating the reduction of the anti-aligned feature (smaller parameters lead to slower decay), resulting in the aligned feature to dominate. **(b):** With a smaller target scale (or larger initialization), minimal adjustments per feature are sufficient to fit the target (lazy).**(c):** Under the NTK initialization with numerous features, gradients toward the target are distributed across the features, resulting in negligible movement for each feature to fit the target (lazy). See Appendix B for details of the illustration.

$\{1, \cdots, p\}$. This gives

$$\sum_{i \in \mathcal{I}_\pm} \frac{da_i^2}{dt} = \mp \frac{2}{Z}(\theta - S) \sum_{i \in \mathcal{I}_\pm} a_i^2,$$

where we used the conserved quantity $a_i^2 - b_i^2 = 0$ and expressed $\theta := Z^{-1} \sum_{i=1}^{p} a_i b_i$ so that $f(x) = x\theta$. Letting $\Delta\theta := Z^{-1} \left( \sum_{i \in \mathcal{I}_+} a_i b_i - \sum_{j \in \mathcal{I}_-} a_j b_j \right)$, the above equations can be rearranged into

$$\frac{d\theta}{dt} = -2\frac{\Delta\theta}{Z}(\theta - S), \quad (16)$$

$$\frac{d(\Delta\theta)}{dt} = -2\frac{\theta}{Z}(\theta - S). \quad (17)$$

If the right hand side of Equation (17) is 0 or $\Delta\theta$ is constant, Equation (16) reduces to linear dynamics (Equation (7)).

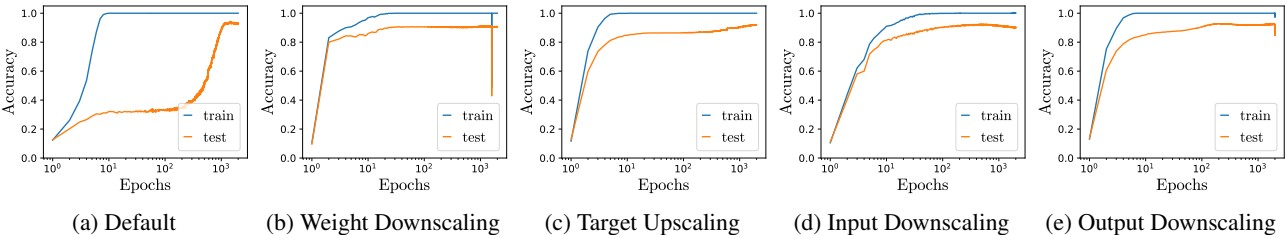

*Figure 8.* **Removing (accelerating) grokking in a Multilayer Perceptron (MLP).** A 4-layer tanh MLP with large weight initialization, trained on 1000 MNIST samples, exhibits grokking (a). All techniques that reduce the weight-to-target ratio $\Sigma_0/S$ (b–e), described in Section 7, place the model in the rich regime and eliminate the generalization delay. See Appendix G.3 for setup details.

Thus, variations in $\Delta\theta$ reflect the differences in the learning speed of aligned and anti-aligned modes. In the rich (greedy) setup, the aligned mode greedily saturate before the anti-aligned mode (Figure 7(a)), while in the lazy (NTK initialization and target downscale) setup, both modes evolve infinitesimally.

**Target downscaling** Target downscaling assumes $Z = 1$, $S \to 0$, and $\theta(0) = 0$ (or $p$ even). Then $\frac{d\Delta\theta}{dt} \approx 0$ as both $\theta$ and $S$ are close to 0. In this case, all features barely need to move to fit the target (Figure 7(b)), avoiding greedy dynamics where a few features dominate. For more rigorous arguments, see Chizat et al. (2019).

**NTK initialization** The NTK limit requires $p \to \infty$ with $Z = \sqrt{p}$. Then $\Delta\theta/Z = \mathcal{O}(1)$ is fixed with $\frac{d\Delta\theta}{dt} \approx 0$ as both $\theta$ and $S$ are of $\mathcal{O}(1)$ as $p \to \infty$. Intuitively, $\Delta\theta$ is fixed because the dynamics spread the required movement $\theta - S = \mathcal{O}(1)$ among $p$ features, resulting in minimal change for individual features. To be specific, each feature moves $\mathcal{O}(p^{-1/2})$, and the collective change of $p$ features sums to $\mathcal{O}(1)$ change in $\theta$. See Figure 7(c) for illustration and Luo et al. (2021) for more rigorous arguments.

**General initialization** In Appendix H, we solve the exact dynamics of Equation (15) on general initialization of $a$ and $b$ (Theorem 3). The analogue of $\Delta\theta$ or the deviation among the features are captured in the following quantity

$$\gamma_+ = \frac{S + \sqrt{\Sigma_0^2 - \mathcal{S}_0^2 + S^2}}{\Sigma_0 + \mathcal{S}_0},$$

where

$$\mathcal{S}_0 = \sum_{i=1}^{p} \frac{a_i(0)b_i(0)}{Z}, \quad \Sigma_0 = \sum_{i=1}^{p} \frac{a_i(0)^2 + b_i(0)^2}{2Z}.$$

The constant $\gamma_+$, loosely speaking, determines how $a_i$ and $b_i$ scale with their initial values. Explicitly, the final limiting value of $a_i(\infty)b_i(\infty)$ is given as

$$\left(\frac{a_i(0) + b_i(0)}{2}\right)^2 \gamma_+ - \left(\frac{a_i(0) - b_i(0)}{2}\right)^2 \gamma_+^{-1}.$$

For $S > S_0$, a large $\gamma_+$ indicates a non-trivial changes in parameters with greater difference in contributions between more and less aligned features — the rich dynamics. However, $\gamma_+ \approx 1$ implies trivial dynamics with negligible changes in $a_i$ and $b_i$, resulting in lazy dynamics. See Appendix H for further discussion.

Note that the ratio between $S$ (target scale) and $\Sigma_0$ (norm at initialization) determines $\gamma_+$ (Remark 5). The small initialization in Sections 4 and 5 maps to small $\Sigma_0/S$, while the NTK initialization (by infinite width) and target downscaling (by small $S$) map to large $\Sigma_0/S$.

### 7.2. Application: Controlling grokking with weight-to-target ratio

As discussed in Section 6, grokking is a transition from the lazy (overfitting) regime to the rich (generalizing) regime. Instead of waiting for the rich regime through overtraining, we can set the model in the rich regime from the initialization. Kumar et al. (2024) empirically verified that the methods to escape the lazy regime (decreasing $\Sigma_0/S$) in our toy model also eliminate the delay in grokking in various settings.

Several methods decrease $\Sigma_0/S$, which coincides with the transition out of the lazy regime: we can increase $S$ by upscaling the target or analogously downscaling the input features (as $S$ is the ratio between $y$ and $x$). Alternatively, we can decrease $\Sigma_0$ by downscaling initialization (Jeffares et al., 2024) or by downscaling the output of the function by increasing $Z$. See Figure 8 or Kumar et al. (2024) for empirical verification and a summary of methods that remove grokking.

**Intuition** For greedy dynamics to emerge, the target must be sufficiently distant from initial output function for better and worse features (modes) to form a gap. Reducing the weight-to-target ratio widens this gap, promoting greedy dynamics that generalize. Based on this understanding, we can interpret grokking as a special case of a poorly initialized setup in the lazy regime.

## 8. Discussion

We demonstrated how the feedback principle unifies seemingly unrelated phenomena as consequences of dynamics of the layerwise structure. In Sections 4 and 5, we explained emergence, scaling laws, and neural collapse through the amplifying dynamics from feedback principle. In Sections 6 and 7, we linked the lazy regime and grokking to the lack of amplification. Throughout, layerwise linear models with specific conditions served as the formal bridge between phenomenon and intuition.

While our investigation mainly focused on layerwise linear models and four phenomena, similar models have also been applied to understand other phenomena, including various architectures (Pinson et al., 2023; Heij et al., 2007; Ahn et al., 2024; Katharopoulos et al., 2020), self supervised learning (Simon et al., 2023), modularization (Jarvis et al., 2023; Zhang et al., 2024a), specialization (Jarvis et al., 2024), fine-tunning (Lippl & Lindsey, 2024), generalization (Advani et al., 2020), and neural representation in neuroscience (Saxe et al., 2019; Richards et al., 2019; Nelli et al., 2023; Flesch et al., 2022; Farrell et al., 2023; Lippl et al., 2024).

## 9. Alternative Views

While layerwise linear models provide valuable insights, we acknowledge their limitations. These models are not used in practice, as DNNs lose expressiveness and performance without non-linearities. Some phenomena may require non-linear activations for explanation, making layerwise linear models too simple. However, given their success in explaining various phenomena — such as grokking which was initially thought to be linked to complicated nature of algorithmic datasets (Power et al., 2022) — the limits of layerwise linear models have yet to be fully explored.

## 10. Conclusion

The success of feedback principle in describing seemingly unrelated phenomena highlights that layerwise structure, even without non-linear activations, is a defining characteristic of DNNs. The simplicity and solvability of this approach in determining whether DNN phenomena result from dynamics make the approach highly appealing.

While layerwise linear models do not offer a complete theory of deep learning, they mark a critical milestone by clarifying how dynamical phenomena emerge from layerwise architecture. They lay the groundwork for deeper intuition and more comprehensive frameworks, helping avoid potential overcomplications. Thus, in this position paper, we argue that layerwise linear models must be solved first to understand a dynamical phenomenon in DNNs.

### 10.1. Call to action: Research directions

**The role of depth** As explored in (Kunin et al., 2024), initialization can be controlled to concentrate feature learning in a few layers or distribute it across many. The former effectively reduces the depth, while the latter fully leverages it. When do these approaches differ in generalization, and why? If performances are similar, what is the true role of depth — purely dynamical or something more fundamental? See for example, Lyu & Zhu (2023) and Wang & Jacot (2024).

**Can we incorporate non-linearity as perturbation?** Physicists often extend simple and solvable systems with perturbative methods to describe more complex systems (Sulejmanpasic & Ünsal, 2018). Since ReLU is partially linear, can we similarly integrate non-linearity into layerwise linear models as a perturbation? Several studies have laid the groundwork in this direction, highlighting the need for continued exploration (Saxe et al., 2022; Kunin et al., 2024; Zhang et al., 2024b).

**The role of batch normalization** Batch normalization (Ioffe & Szegedy, 2015) has an interesting connection to the setup of layerwise linear models. For a given layer, it normalizes the features and introduces a learnable parameter — $\gamma^{(k)}$ in Ioffe & Szegedy (2015) — that scales the input. The Gaussianized inputs align with the whitening assumption in linear neural networks, while the scaling parameter increases similarity to diagonal linear neural networks. Further research in this direction may provide a connection between batch normalization and the rich regime.

**The role of skip connections** Skip connections introduce varying degrees of multilinearity across layers, potentially inducing lazy dynamics in some and rich dynamics in others, thereby assigning distinct roles to each layer. A deeper dynamical understanding of skip connections may clarify the roles of layers at different depths.

**The role of optimizers** The role of optimizers in shaping amplifying dynamics — such as whether they enhance or regulate greedy behavior — remains underexplored. While the inductive bias of stochastic gradient descent has been studied (Lyu & Zhu, 2023), the impact of second-order methods is largely unknown. A solvable model could provide clearer insights into the benefits and limitations of these methods in the context of rich dynamics.

For additional research directions, see Appendix I.

## Acknowledgment

We thank Nayara Fonseca de Sá, Sewook Oh, Zhanxing Zhu, and Bochen Lyu for helpful discussions. SL was supported by the National Research Foundation of Korea (NRF) grants No.2020R1A5A1016126 and No.RS–2024-00462910. CD was supported by the Gatsby Charitable Foundation (GAT3755). This research was funded in whole, or in part, by the Wellcome Trust [216386/Z/19/Z]. YP is supported by the faculty research fund of Sejong University in 2025. CL was supported by the Engineering and Physical Sciences Research Council grant EP/W524311/1. This research was funded in whole or in part by the Wellcome Trust [216386/Z/19/Z]. For the purpose of Open Access, the author has applied a CC BY public copyright licence to any Author Accepted Manuscript version arising from this submission. The authors would also acknowledge support from His Majesty's Government in the development of this research.

## Impact Statement

This position paper advocates for research directions based on a dynamic principle and intuitively explains phenomena of DNN. Therefore, we assess no societal impact that requires attention.

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

# A. Glossary

| Name | Symbol /Abbr. | Definition | Cross-ref |
|---|---|---|---|
| Amplifying dynamics | | The rich-get-richer dynamics of layerwise linear models stemming from the dynamical feedback principle. | Section 4 |
| Dimension of input features | $d$ | | |
| Dimension of hidden layer | $p$ | | |
| Dimension of output | $c$ | | |
| Dynamical feedback principle | | The principle that magnitude of one layer governs the evolution of the other layers. | Section 3 |
| Emergence | | A phenomenon where DNN suddenly gains performance on a subtask or skill. | Appendix D |
| Gradient flow | GF | A continuous variant of gradient descent algorithm | |
| Greedy dynamics | | The dynamics of low-rank bias. | Section 5 |
| Grokking | | A delayed generalization. | Appendix G |
| Input features | $x$ | $x \in \mathbb{R}^d$. We notate the element as $x_i$ and the input feature space with $\mathcal{X}$. | Section 2 |
| Layerwise linear models | | Two layer models in which the output is multilinear with respect to each layer of parameters. | Appendix B |
| Lazy regime | | Regime where DNN's evolution follows a linear dynamics | Section 6 |
| Linear dynamics | | The dynamics of linear model | Equation (7) |
| Linear model | | One layer model in which the output is linear with respect to parameters. | Appendix B |
| Low-rank bias | | An inductive bias of linear neural networks toward low-rank representation. | Section 5 |
| Mode | | A distinct degree of freedom the dynamical system decoupled into. | Appendix B |
| Multilayer perceptron | MLP | Also known as fully connected network (FCN) | |
| Neural collapse | NC | The collapse of training set's last layer features vectors into a low-rank structure. | Appendix E |
| Normalization constant | Z | The constant that rescales the output of the model. | Equation (15) |
| Neural tangent kernel | NTK | kernel with the gradient operator as the feature map. | |
| Parameters (diagonal linear neural network) | $a, b$ | $a, b \in \mathbb{R}^d$ $a_i, b_i$ are used to indicate the $i^{\text{th}}$ entry. In Section 7, the notation $a, b \in \mathbb{R}^p$ was duplicately used for the parameters of linear neural networks for demonstrative purposes. | Equation (3) |
| Parameters (linear neural network) | $W_1, W_2$ | $W_1 \in \mathbb{R}^{d \times p}$, $W_2 \in \mathbb{R}^{p \times c}$. $W_{ij}^{(1)}, W_{ij}^{(2)}$ are used to indicate the element of matrix. | Equation (4) |
| Rich regime | | Regime where DNN's evolution follows a non-linear dynamics | Section 6 |
| Sigmoidal growth | | The evolution amplifying dynamics when $a, b \ll S$. | Equation (9) |
| Skill | $g_k(x)$ | A subtask of the problem. | Section 4 |
| Stage-like training | | The dynamics when modes are learned in sequences without overlap in their saturation. | Figure 3(b) |
| Target (scale) | $S_i$ or $S$ | The multiplicative constant to the feature in the target function. | Equation (1) |
| Weight-to-target ratio | $\Sigma_0 / S$ | The ratio between the normalized (by $Z$) norm of initial weights and the target scale $S$. | Section 7 |

# B. Definitions

In this section, we provide more detailed definitions of the following terms: linear models, layerwise linear models, modes, features, function space. We additionally explain Figure 7. We assume scalar output ($c = 1$) and focus on models with $f : \mathcal{X} \to \mathbb{R}$ where $\mathcal{X}$ is the input feature space.

## B.1. Linear models

Linear models are models in which the output is completely linear in **parameters**. The definition allows non-linear relationship between the input features and the output. The general form of a linear model is

$$f(x) = \sum_{i=1}^{d} \phi_i(x)\theta_i, \tag{18}$$

where $\phi_i : \mathcal{X} \to \mathbb{R}$ is a feature map that can be a non-linear function.

## B.2. Layerwise linear models

Layerwise linear models are 2-layer models in which the output is multilinear with respect to each layer of parameters. The definition also allows non-linear relationship between the input features and the output. The general form of a layerwise linear model is

$$f(x) = \sum_{(i,j)\in\mathcal{V}} \sum_{j=1}^{p} \phi_i(x)a_{ij}b_j, \tag{19}$$

where $\phi : \mathcal{X} \to \mathbb{R}$ is a map from the input data space $\mathcal{X}$ to the reals, and $\mathcal{V}$ is an architecture dependent set of connections between $\phi_i(x)$ and hidden neurons. Note that the presence of $\phi$ in place of $x_i$ does not change the dynamics, as long as random variable $\phi_i(x)$ satisfy the assumptions in the main text, such as $\mathbf{E}[\phi_i(x)\phi_j(x)] = 0$ for $i \neq j$.

## B.3. Modes

For a given dynamics, we define modes as distinct degrees of freedom into which the dynamical system decouples.

## B.4. Features

We return to the special case of $\phi_i(x) = x_i$ to maintain greater consistency with the main text. For the last layer feature vectors used in Section 5, see Appendix E instead.

We define the $k^{\text{th}}$ feature as the post-activation of the $k^{\text{th}}$ neuron of the last layer multiplied by the respective last layer parameter. For a linear model, the $k^{\text{th}}$ feature is

$$h_k := x_k\theta_k, \qquad k \in \{1, 2, \cdots, d\}. \tag{20}$$

For a linear neural network, the $k^{\text{th}}$ feature is

$$h_k := \sum_{i=1}^{d} x_i W_{ik}^{(1)} b_k, \qquad k \in \{1, 2, \cdots, p\}. \tag{21}$$

Note that by the definition, the sum of all features equals the output function:

$$f = \sum_{k} h_k. \tag{22}$$

## B.5. Function space and the explanation of Figure. 7

We define the **function space** as the vector space spanned by $x_1, \cdots, x_d$: that is, we only consider linear functionals in variables $x_1, \cdots, x_d$ among all functionals on $\mathcal{X}$. Because $x$ is a $d$ dimensional random variable with full ranked covariance matrix, $x_1, \cdots, x_d$ are linearly independent and thus form the basis of function space.

$d = 2, p = 2, S \gg 1$

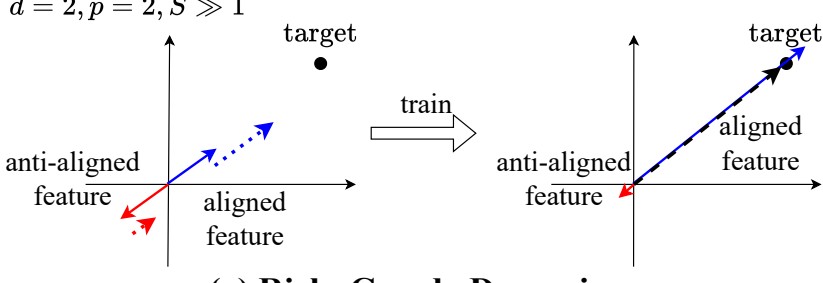

### (a) Rich: Greedy Dynamics

$d = 2, p = 2, S \ll 1$

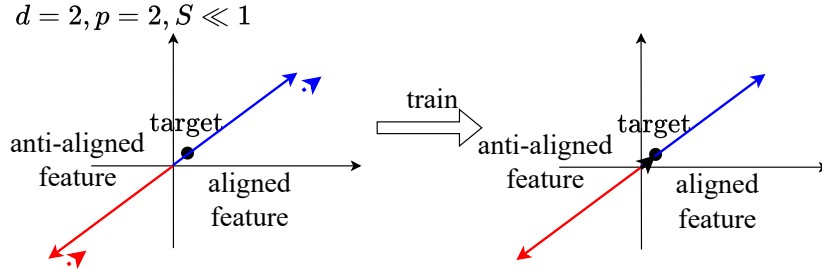

### (b) Lazy: Target Downscaling

$d = 2, p \gg 1, S = \mathcal{O}(1)$

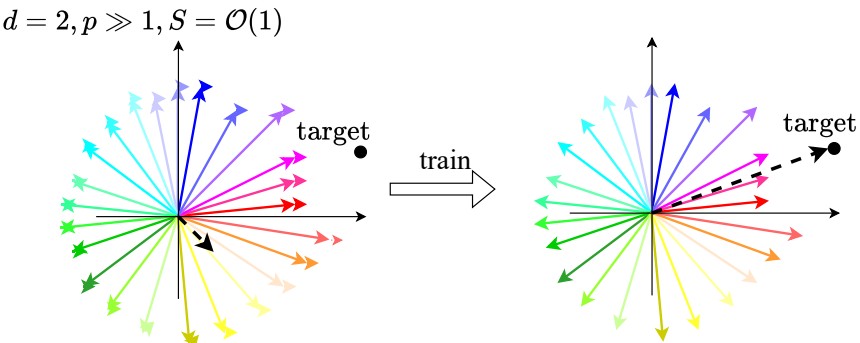

### (c) Lazy: NTK Initialization

*Figure 9.* **Figure 7 in the main text.**

Given our orthogonality condition of features ($\mathbf{E}[x_i x_j] = 0$ for $i \neq j$), we can use the input feature as the (unnormalized) basis, and express the target function $f^*(x) = \sum_i^d x_i S_i$ as a vector in this function space with coordinate $[S_1, S_2, \cdots, S_d]$. Likewise, the features can be expressed as a vector in the function space, allowing visualization.

In Figure 9(a), we have two axis because $d = 2$ ($[x_1, x_2]$), and two features because $p = 2$ (Equation (21)). The blue feature is aligned ($\mathbf{E}[h^T f^*] > 0$) and the red feature is anti-aligned ($\mathbf{E}[h^T f^*] < 0$) to the target, analogous to the aligned and anti-aligned features in the $d = 1$ setup of the main text (Equation (15)). The sum of features equals the current output function (Equation (22)) or the black dashed line. The training ends when the output function equals the target function (black dot).

In Figure 9(c), we observe $p \gg 1$ features (colored arrows) sum to a smaller output function (dashed black line). A collection of infinitesimal changes in these features can sum up to an $\mathcal{O}(1)$ change in expressing the target function.

# C. Derivation for completeness

This section includes simple derivations for completeness. For the original derivations in a more general setup, we refer to the references mentioned in the main text.

## C.1. Dynamics of linear model

The gradient flow equation is

$$\frac{d\theta_i}{dt} = -\frac{\partial \mathcal{L}}{\partial \theta_i}, \tag{23}$$

where $\mathcal{L}$ is the MSE loss (Equation (2)). Then

$$\frac{d\theta_i}{dt} = -\mathbf{E}\left[\frac{\partial}{\partial \theta_i} \frac{1}{2}(f(x) - f^*(x))^2\right] \tag{24}$$

$$= -\mathbf{E}\left[x_i \left(\sum_{j=1}^{p}(\theta_j - S_j)x_j\right)\right] \tag{25}$$

$$= -\mathbf{E}\left[x_i^2\right](\theta_i - S_i). \tag{26}$$

In the second line, we used the definition of the linear model (Equation (5)) and the target function (Equation (1)). In the last line, we used our assumptions that the input features are orthogonal ($\mathbf{E}[x_i x_j] = 0$ if $i \neq j$) and that we have an infinite training set (the expectation is over the true distribution).

Solving Equation (26), we obtain

$$\log(S_i - \theta_i(t)) - \log(S_i - \theta(0)) = -\mathbf{E}\left[x_i^2\right]t. \tag{27}$$

Using the zero initialization condition of $\theta(0) = 0$, we obtain Equation (10):

$$\theta_i/S_i = 1 - e^{-\mathbf{E}[x_i^2]t}. \tag{28}$$

## C.2. Dynamics of diagonal linear neural network

Starting with the gradient flow equation in diagonal linear neural networks,

$$\frac{da_i}{dt} = -\frac{\partial \mathcal{L}}{\partial a_i}, \quad \frac{db_i}{dt} = -\frac{\partial \mathcal{L}}{\partial b_i}. \tag{29}$$

Analogous to Equation (26), we can use the orthogonality condition ($\mathbf{E}[x_i x_j] = 0$ if $i \neq j$) to get

$$\frac{da_i}{dt} = -\mathbf{E}\left[\frac{\partial}{\partial a_i} \frac{1}{2}(f(x) - f^*(x))^2\right] \tag{30}$$

$$= -\mathbf{E}\left[b_i x_i \left(\sum_{j=1}^{p}(a_j b_j - S_j)x_j\right)\right] \tag{31}$$

$$= -b_i \mathbf{E}\left[x_i^2\right](a_i b_i - S_i). \tag{32}$$

We can analogously obtain $\frac{db_i}{dt}$, and the evolution of $a_i b_i$ is

$$\frac{d(a_i b_i)}{dt} = \frac{da_i}{dt} b_i + a_i \frac{db_i}{dt} \tag{33}$$

$$= -\mathbf{E}\left[x_i^2\right](b_i^2 + a_i^2)(a_i b_i - S) \tag{34}$$

$$= -2\mathbf{E}\left[x_i^2\right] a_i b_i(a_i b_i - S), \tag{35}$$

where we used Equation (32) (and its equivalent for $b_i$) in the second line and the condition $a_i = b_i$ in the last line. Assuming $a_i(0)b_i(0) < S$, we can solve the differential equation to obtain

$$a_i(t)b_i(t) = \frac{S_i}{1 + \left(\frac{S_i}{a_i(0)b_i(0)} - 1\right)e^{-2S_i\mathbf{E}[x_i^2]t}}. \tag{36}$$

For a general derivation when $a_i \neq b_i$, see Appendix A of Saxe et al. (2014).

### C.3. Derivation of the magnitude difference conservation in linear neural network

For the sake of readability, we restate the dimensions of inputs, weights, and outputs (labels) for linear neural networks:

$$x \in \mathbb{R}^{d\times 1}, \quad y \in \mathbb{R}^{c\times 1}, \quad W_1 \in \mathbb{R}^{d\times p}, \quad W_2 \in \mathbb{R}^{p\times c}. \tag{37}$$

For a linear neural network, the gradient flow equation is

$$\frac{dW_1}{dt} = -\frac{\partial \mathcal{L}}{\partial W_1}, \quad \frac{dW_2}{dt} = -\frac{\partial \mathcal{L}}{\partial W_2}. \tag{38}$$

The loss in the summand notation is

$$\mathcal{L} = \frac{1}{2}\mathbf{E}\left[\sum_k \left(\sum_{i,j} x_i w_{ij}^{(1)} w_{jk}^{(2)} - y_k\right)\left(\sum_{i',j'} x_{i'} w_{i'j'}^{(1)} w_{j'k}^{(2)} - y_k\right)\right], \tag{39}$$

where $w_{ij}^{(1)}$ and $w_{jk}^{(2)}$ are elements of matrix $W_1$ and $W_2$, respectively. The derivative for the first matrix $W_1$ is

$$\frac{\partial \mathcal{L}}{\partial w_{ij}^{(1)}} = \mathbf{E}\left[\sum_k \left(x_i w_{jk}^{(2)}\right)\left(\sum_{i',j'} x_{i'} w_{i'j'}^{(1)} w_{j'k}^{(2)} - y_k\right)\right] \tag{40}$$

$$= \mathbf{E}\left[x_i \left(\sum_{i',j',k} x_{i'} w_{i'j'}^{(1)} w_{j'k}^{(2)} w_{jk}^{(2)} - y_k w_{jk}^{(2)}\right)\right] \tag{41}$$

and this can be expressed in the matrix form as

$$\frac{\partial \mathcal{L}}{\partial W_1} = \mathbf{E}\left[x(x^T W_1 W_2 - y^T)W_2^T\right]. \tag{42}$$

Likewise for $W_2$,

$$\frac{\partial \mathcal{L}}{\partial w_{jk}^{(2)}} = \mathbf{E}\left[\left(\sum_i x_i w_{ij}^{(1)}\right)\left(\sum_{i',j'} x_{i'} w_{i'j'}^{(1)} w_{j'k}^{(2)} - y_k\right)\right], \tag{43}$$

which in the matrix form is

$$\frac{\partial \mathcal{L}}{\partial W_2} = \mathbf{E}\left[W_1^T x(x^T W_1 W_2 - y^T)\right]. \tag{44}$$

Using Equations (38), (42) and (44), we obtain

$$W_1^T \frac{dW_1}{dt} = \frac{dW_2}{dt}W_2^T \quad \Rightarrow \quad \frac{d}{dt}(W_1^T W_1 - W_2 W_2^T) = 0. \tag{45}$$

### C.4. Derivation of Equation (12)

Here, we show that a special initialization of $W_1$ and $W_2$ leads to Equation (12) and argue that a sufficiently small initialization makes the assumption plausible. For a more rigorous arguments, see (Saxe et al., 2014; Mixon et al., 2020; Ji

& Telgarsky, 2019; Arora et al., 2019a; Lampinen & Ganguli, 2019; Gidel et al., 2019; Tarmoun et al., 2021; Braun et al., 2022; Dominé et al., 2025), especially more recent works for formal handling of initialization.

Let the singular value decomposition of the correlation matrix be

$$U^T P V = \mathbf{E}\left[xy^T\right]. \tag{46}$$

We assume whitened input $\mathbf{E}[xx^T] = I$, $W_1(0) = U^T A R$, and $W_2(0) = R^T A V$. The diagonal matrix $A$ is of rank $c$ and comprises sufficiently small singular values, while $R \in \mathbb{R}^{c \times c}$ is an orthogonal matrix. These assumptions on $W_1$ and $W_2$ immediately shows that layers are balanced:

$$W_1^T W_1 - W_2 W_2^T = 0. \tag{47}$$

Denote

$$\widetilde{W}_1 = U W_1 R^T, \quad \widetilde{W}_2 = R W_2 V^T. \tag{48}$$

Then from the gradient dynamics

$$\frac{dW_1}{dt} = -\mathbf{E}[xx^T]W_1 W_2 W_2^T + \mathbf{E}[xy^T]W_2^T = -W_1 W_2 W_2^T + U^T P V W_2^T, \tag{49}$$

we have

$$\frac{d\widetilde{W}_1}{dt} = \frac{d(U W_1 R^T)}{dt} = -(U W_1 R^T)(R W_2 V^T)(V W_2^T R^T) + (U U^T)P(V W_2^T R^T) \tag{50}$$

$$= -\widetilde{W}_1 \widetilde{W}_2 \widetilde{W}_2^T + P \widetilde{W}_2^T. \tag{51}$$

Similarly, for $\widetilde{W}_2$,

$$\frac{d\widetilde{W}_2}{dt} = -\widetilde{W}_1^T \widetilde{W}_1 \widetilde{W}_2 + \widetilde{W}_1^T P. \tag{52}$$

As both $\widetilde{W}_1$ and $\widetilde{W}_2$ starts with diagonal matrix $\widetilde{W}_1(0) = \widetilde{W}_2(0) = A$, and all matrices in Equation (51) and Equation (52) are diagonal, $\widetilde{W}_1$ and $\widetilde{W}_2$ are always diagonal. Also from

$$\widetilde{W}_1^T \widetilde{W}_1 - \widetilde{W}_2 \widetilde{W}_2^T = R(W_1^T W_1 - W_2 W_2^T)R^T = 0, \tag{53}$$

we have $\widetilde{W}_1(t) = \widetilde{W}_2(t)$ for any $t$. Denote by $\alpha_i$ the $i^{\text{th}}$ diagonal entry of $\widetilde{W}_1$. We have either from Equation (51) or Equation (52),

$$\frac{d\alpha_i}{dt} = -\alpha_i^3 + \rho_i \alpha_i, \tag{54}$$

where $\rho_i$ is the $i^{\text{th}}$ diagonal entry of $P$ or the $i^{\text{th}}$ singular value of the correlation matrix. Solving this equation gives

$$\frac{d\alpha_i^2}{dt} = -2\alpha_i^2(\alpha_i^2 - \rho_i) \quad \Rightarrow \quad \frac{\alpha_i^2(t)}{\rho_i} = \frac{1}{1 + \left(\frac{\rho_i}{\alpha_i^2(0)} - 1\right)e^{-2\rho_i t}}, \tag{55}$$

and the dynamics of $W_1 W_2$ is given as

$$W_1 W_2 = U^T A(t)^2 V, \quad A(t)^2 = \begin{pmatrix} \alpha_1(t)^2 & & \\ & \alpha_2(t)^2 & \\ & & \ddots \end{pmatrix}. \tag{56}$$

**Discussion on the assumption**  The assumption specifies a particular form for the matrix, but with sufficiently small initialization, all matrices approximately meet the conditions.

# D. Emergence and scaling laws

In this section, we review the emergence and (neural) scaling laws in the literature and discuss in detail how sigmoidal dynamics explain emergence, particularly in relation to data and parameters.

## D.1. Scaling law

Neural scaling laws refer to the empirical observation that the performance of DNNs follows a power-law relationship with the number of parameters or the amount of data (Hestness et al., 2017; Kaplan et al., 2020). This phenomenon has been observed across various architectures and datasets (Rosenfeld et al., 2020; Henighan et al., 2020; Gordon et al., 2021; Zhai et al., 2022; Hoffmann et al., 2022; Cabannes et al., 2024; Bachmann et al., 2024).

To explain the data scaling law, Hutter (2021) developed a discrete feature model where the frequency of features follows a power-law, leading to a power-law relationship between performance and the number of datapoints. Similar assumptions have been used in recent works, often using linear models such as kernels, that a power-law in input distribution can explain scaling laws (Spigler et al., 2020; Bordelon et al., 2020; Cui et al., 2021; Bahri et al., 2021; Maloney et al., 2022; Bordelon et al., 2024).

## D.2. Emergence

In machine learning, emergence refers to a sudden improvement in DNN's performance when a key factor (such as the size of the training set or the number of parameters) increases (Ganguli et al., 2022; Srivastava et al., 2023; Wei et al., 2022). Such observations are often made in large language models (Brown et al., 2020), where LLMs demonstrate certain abilities, such as logical reasoning, once the model size exceeds a specific threshold. For example, Wei et al. (2022) quoted *"An ability is emergent if it is not present in smaller models but is present in larger models."*

Although a concrete definition of emergence in machine learning remains elusive (Anwar et al., 2024) and has been criticized for its sensitivity to measurement (Schaeffer et al., 2023), many theoretical studies have explored its causes. Emergence has been deemed challenging to explain through *"mathematical analysis of gradient-based training"* (Arora & Goyal, 2023) and is often viewed as a collection of basic skills that interact to suddenly produce a complex ability (Arora & Goyal, 2023; Chen et al., 2023; Yu et al., 2024; Okawa et al., 2024).

## D.3. Scaling laws from emergence

Similar to the approach of Hutter (2021), Michaud et al. (2023) proposed a quanta model and the multitask sparse parity dataset to link emergence (a sudden gain in ability) to the scaling laws. The quanta model posits that a skill is either learned or not, depending on the available resources (time, data, and parameters). The multitask sparse parity problem consists of skills whose frequencies follow a power law. By leveraging the power-law distribution of the skill frequencies, they demonstrated that the sum of quantized learning, in the large limit, leads to scaling laws observed in a 2-layer MLP trained on the multitask sparse parity problem.

## D.4. Emergence from dynamics

As discussed in Section 4, authors of Nam et al. (2024) proposed that emergence arises from dynamics. They used a layerwise linear model with prebuilt skill functions (Equation (11)). The learning of a skill corresponds to the learning of a mode, and the sigmoidal dynamics naturally explain the time emergence (Figure 4(a)). However, the connections to data and parameter emergence (Figure 4(b,c)) were less explained in the main text.

The arguments for data and parameter emergence in Nam et al. (2024) are two-folds: (1) Justifying the multilinear model as a valid approximation of practical DNNs and (2) solving the multilinear model's dynamics for data and parameter emergence.

**Justification of the multilinear model and prebuilt features.** Multilinear models (Equation (3)) are aware of different skills due to the prebuilt skill functions while DNNs do not inherently separate skills. Authors of Nam et al.(2024) justify the behavior through the **layerwise dynamics** (dynamical feedback in our paper): the layerwise structure and the power-law in skill frequency lead to stage-like feature learning where skills emerge in distinct phases and the dynamics of learning the skills remain effectively decoupled. Figure 4(a) shows an example in which neural networks learn different features in a stages.

Intuitively, we can imagine the stage-like feature learning as DNNs prioritizing more frequent skills under resource constraints. In parameter emergence, limited neurons are allocated to express frequent skills only — as frequent skills are learned faster — creating a sharp performance gap between more and less frequent skills. Data emergence is less direct, but more samples from the skill speed up learning of the skill. Dynamic feedback amplifies this gap, enabling DNNs to distinguish more frequent skills, while less frequent skills remain entangled with each other, resulting in significantly lower skill performance.

**Data and parameter emergence in a multilinear model**   In the setup of Nam et al,(2024), Equation (3) has only the first $p$ most frequent (with the largest $\mathbf{E}[g_i(x)^2]$) skill functions out of $p^*$ to fully express the target, and the skill functions are mutually exclusive:

$$f(x; a, b) = \sum_{i=1}^{p} a_i b_i g_i(x), \qquad g_i(x)g_j(x) = 0 \ \text{if} \ i \neq j. \tag{57}$$

The two conditions state that a model will learn the $k^{\text{th}}$ skill if and only if a datapoint from the $k^{\text{th}}$ skill exists in the training set — Corollary 1 of Nam et al.(2024) — and the skill function exists in the model ($p \geq k$) — Corollary 2 of Nam et al.(2024) — creating emergent (abrupt) property of the model on data and parameters.

For data Emergence, Corollary 1 of Nam et al.(2024) states that a multilinear model is a one-shot learner, meaning a single observation of the $k^{th}$ skill, for infinite training time and infinite parameters, leads to a training of the skill. Thus, we have (with $p, t \to \infty$):

$$a_k(\infty)b_k(\infty) = \begin{cases} d_k > 0 : & S \\ d_k = 0 : & \mathcal{R}_k(0) \end{cases} \tag{58}$$

Given a finite training set, the expected probability of observing the $k^{th}$ skill depends on the training set size. Because of the power-law in skill frequency, this value shows a relatively abrupt change of training set size in log-scale, and thus data emergence. In Nam et al.(2024), the authors extended this model to a $D_k$ shot learner instead of a one-shot learner and used the model to fit Figure 4(b).

For parameter Emergence, Corollary 2 of Nam et al.(2024) states that because the skill functions are mutually exclusive, the model will learn the $k^{th}$ skill - for an infinite training set and time - whenever the $k^{th}$ skill function $g_k$ exists in the model. Formally (with $n, t \to \infty$),

$$a_k(\infty)b_k(\infty) = \begin{cases} k \leq p : & S \\ k > p : & \mathcal{R}_k(0) \end{cases} \tag{59}$$

where $p$ is the width of the multilinear model. The parameter emergence is trivial as the model will suddenly learn a skill once corresponding skill function is added. In Nam et al.(2024), the authors extended this model such that $N_k$ hidden neurons instead of one to fit Figure 4(c).

One intuitive connection back to stage-like training of MLP is that all neurons, by time-emergence, will be used to fit the **most frequent skill first**; thus exhausting the limited hidden neurons to express more frequent skills only. This will create an inductive bias toward more frequent skill and justify why a larger width leads to an abrupt learning of a less frequent skill. See Nam et al. (2024) for a further discussion.

# E. Neural collapse

In this section, we restate the four NC conditions given in Papyan et al. (2020), and discuss their significance.

## E.1. Setup

We decompose a neural network, a map from input feature space $\mathcal{X}$ to output $\mathbb{R}^c$, into two components: 1) a map from the input features to the last layer features $\Phi : \mathcal{X} \to \mathbb{R}^p$ and 2) the last layer (a map from $\mathbb{R}^P$ to $\mathbb{R}^c$). Then, we can express the neural network as

$$f_k(x) = \sum_{i=1}^{p} \Phi_i(x) W_{ik} + b_k, \tag{60}$$

where $W$ and $b \in \mathbb{R}^p$ are the weight and bias terms of the last layer. We use size $n$ **training set** that is partitioned into $c$ sets $[A_1, A_2, \cdots, A_c]$ of the same size (balanced classification), where each $A_k$ consists of all training datapoints with true label $k$. The $k^{\text{th}}$ class mean vector is

$$\mu_k := \frac{c}{n} \sum_{x \in A_k} \Phi(x), \tag{61}$$

where $n$ is the number of training set. The global mean vector is

$$\mu_g := \frac{1}{c} \sum_{k=1}^{c} \mu_k. \tag{62}$$

The feature within-class covariance $\Sigma_W \in \mathbb{R}^{p \times p}$ is

$$\Sigma_W := \frac{1}{n} \sum_{k=1}^{c} \sum_{x \in A_k} (\Phi(x) - \mu_k)(\Phi(x) - \mu_k)^T, \tag{63}$$

which shows the deviation of last layer feature vectors away from the corresponding class mean vectors.

## E.2. NC conditions

The four NC conditions are the following:

1. **Within-class variance tends to 0**

$$\Sigma_W \to 0. \tag{64}$$

2. **Convergence to simplex ETF**

$$\frac{(\mu_j - \mu_g)^T (\mu_k - \mu_g)}{\|(\mu_j - \mu_g)\|_2 \|(\mu_k - \mu_g)\|_2} \to \frac{c\delta_{jk} - 1}{c - 1}. \tag{65}$$

3. **Convergence to self duality**

$$\frac{w_k}{\|w_k\|_2} - \frac{\mu_k - \mu_g}{\|(\mu_k - \mu_g)\|_2} \to 0. \tag{66}$$

4. **Simplification to nearest class center**

$$\arg\max_k \left( \sum_{i=1}^{p} \Phi_i(x) W_{ik} + b_k \right) \to \arg\min_k \|\Phi(x) - \mu_k\|_2. \tag{67}$$

Condition 1 states that all feature vectors collapse around their class mean vector. Condition 2 states that the class mean vectors form a simplex ETF structure (Figure 5). Condition 3 states that the last layer found the minimum-norm solution for the last layer feature vectors satisfying conditions 1 and 2. The last condition states that the model assesses classification based on the distance between the last layer feature vectors and the class mean vectors.

# F. Imbalanced layer initialization and data-adaptiveness

We extend the discussion on the topic developed in Section 6. Note that we summarize here information mainly taken from Domine et al. (2025) and Kunin et al. (2024).

## F.1. The balanced condition

Let us formally reintroduce Equation (14):

**Definition 1** (Definition of $\lambda$-*balanced* Property ((Saxe et al., 2014; Marcotte et al., 2024)))**.** The weights $W_1$ and $W_2$ are said to be $\lambda$-*balanced* if and only if there exists a **Balanced Coefficient** $\lambda \in \mathbb{R}$ such that:

$$W_2^T W_2 - W_1 W_1^T = \lambda I.$$

When $\lambda = 0$, this property is referred to as **Zero-Balanced**, and it satisfies the condition:

$$W_2^T W_2 - W_1 W_1^T = 0,$$

This condition is of interest because it remains conserved for any initial value $\lambda$, as detailed in (Marcotte et al., 2024; Dominé et al., 2025). It becomes particularly relevant in the continual learning, reversal learning, and transfer learning settings.

This quantity can be interpreted as the relative scale across layers. A straightforward intuition for the relative scale, as compared to the absolute scale (the norm $W_2 W_1$ when target scale is fixed to 1), can be gained by considering the scalar case where the input, output and hidden dimension are equal to 1. In this simple scenario, it is easy to ensure that $w_1^2 = w_2^2$ satisfies $\lambda = 0$ while allowing for different absolute scales. For example, $w_1 = w_2 = 0.001$ or $w_1 = w_2 = 5$. In such cases, the absolute scale is clearly decoupled from the relative scale. However, in more complex settings, the interaction between relative and absolute scales becomes non-trivial. As we will discussed below, their role in shaping and amplifying dynamics is intricate and highly dependent on the underlying network architecture.

**Intuition** Even with non-linearity, one layer's weight controls the other's rate of change (Section 3). This principle enables targeted training of a layer, allowing tuning between lazy and rich (and thus grokking) dynamics. This intuition can be formalized as

**Theorem 2.** *Under the assumptions of whitened inputs (1), $\lambda$-balanced weights (2), and no bottleneck and with a task-aligned initialization, as defined in (Saxe et al., 2014), the network function is given by the expression $W_2 W_1(t) = \tilde{U} P(t) \tilde{V}^T$ where $P(t) \in \mathbb{R}^{c \times c}$ is a diagonal matrix of singular values with elements $\rho_\alpha(t)$ that evolve according to the equation,*

$$\rho_\alpha(t) = \rho_\alpha(0) + \gamma_\alpha(t; \lambda) \left( \tilde{\rho}_\alpha - \rho_\alpha(0) \right), \tag{68}$$

*where $\tilde{\rho}_\alpha$ is the $\alpha$ singular value of the correlation matrix and $\gamma_\alpha(t; \lambda)$ is a $\lambda$-dependent monotonic transition function for each singular value that increases from $\gamma_\alpha(0; \lambda) = 0$ to $\lim_{t \to \infty} \gamma_\alpha(t; \lambda) = 1$ defined explicitly in Appendix of (Dominé et al., 2025). We find that under different limits of $\lambda$, the transition function converges pointwise to the sigmoidal ($\lambda \to 0$) and exponential ($\lambda \to \pm\infty$) transition functions,*

$$\lim_{\lambda \to 0} \gamma_\alpha(t; \lambda) = \frac{e^{2\tilde{s}_\alpha \frac{t}{\tau}} - 1}{e^{2\tilde{s}_\alpha \frac{t}{\tau}} - 1 + \frac{\tilde{s}_\alpha}{s_\alpha(0)}}, \qquad \lim_{\lambda \to \pm\infty} \gamma_\alpha(t; \lambda) = 1 - e^{-|\lambda| \frac{t}{\tau}}. \tag{69}$$

The proof for Theorem 2 can be found in (Dominé et al., 2025). As $\lambda$ increases, the dynamics of the network resembles those of a linear model, transitioning into the *lazy* regime. On the other end, as lambda goes to zero the networks learns the most salient features first, which can be beneficial for generalization (Lampinen & Ganguli, 2019).

## F.2. Controlling the dynamics with layer imbalance

We conducted an experiment to explore the relationship between relative weight scale, absolute weight scale, and the network's learning regime in a general setting as shown in Figure 6. The absolute scale of the weights in Figure 6(a) is defined as the norm of $W_2 W_1$. Random initial weights with specified relative and absolute scales were generated, and the network was trained on a random input-output task. During training, we calculated the logarithmic kernel distance of the

NTK from initialization and the logarithmic loss. The kernel distance is calculated as: $S(t) = 1 - \frac{\langle K_0, K_t \rangle}{\|K_0\|_F \|K_t\|_F}$. as defined in Fort et al. (2020). These values were visualized as heat maps for $\lambda \in [-9, 9]$ and relative scales in $(0, 20]$. The regression task parameters were set with $(\sigma = \sqrt{3})$. The task has batch size $N = 10$. The network has with a learning rate of $\eta = 0.01$. The lambda-balanced network are initialized with $\mathbf{E}[xy^T] = I$ of a random regression task. Figure 6(a) shows that a square $(d = c)$ linear neural network satisfying Equation (14) only show amplifying dynamics when the weights are balanced with small $|\lambda|$. Larger imbalances lead to linear dynamics in the training of a single layer. However, as the relationship becomes more complex.(Dominé et al., 2025) demonstrates that both relative and absolute scales significantly impact the learning regime of the network across different architectural types.

Note that here, we examine how much the NTK has shifted by the end of training, without focusing on the dynamics of the NTK throughout the training process. As discribed above, lazy training is associated with exponential dynamics, while rich training is linked to sigmoidal dynamics. As highlighted in Braun et al. (2022), the dynamical transition in the structure of internal representations can be decoupled. This result demonstrates that rich, task-specific representations can emerge at any scale when $\lambda = 0$, with the dynamics transitioning from sigmoidal to exponential as the scale increases.

### F.3. Controlling grokking with layer imbalance

As discussed above, larger weight imbalance (in any direction) leads the transition from the rich to the lazy regime in square linear neural networks (Figure 6). For neural networks with non-linear activations, the non-linear activations break the symmetry between layers, potentially initiating feature learning for $\lambda \to \infty$ (Kunin et al., 2024). Here, we explore how the **architecture** can break the symmetry and lead to grokking — a transition from the lazy to rich regime — in linear neural networks, followed by its application to networks with non-linear activations.

**Grokking in linear neural networks**   We introduce two networks: **funnel networks** that narrow from input to output, and **anti-funnel networks** that expand from input to output. In Figure 10, we observe that funnel networks enters the lazy regime as $\lambda \to -\infty$ (downstream initialization), while anti-funnel networks do so as $\lambda \to \infty$ (upstream initialization), which are consistent with Figure 6. However, the networks in opposite limits ($\lambda \to \infty$ for Figure 10(a) and $\lambda \to -\infty$ for Figure 10(b)) show significant change in the NTK after training (rich regime). The networks were initially at the lazy regime, but showed a transition to a delayed rich regime (grokking).

Intuitively, the gradient from lighter layer negligibly changes the heavier layer (lazy), but the wider input (output) of funnel (anti-funnel) networks allows the lighter layer to aggregate sufficient gradients on the heavier layer, driving a meaningful change into a rich regime.

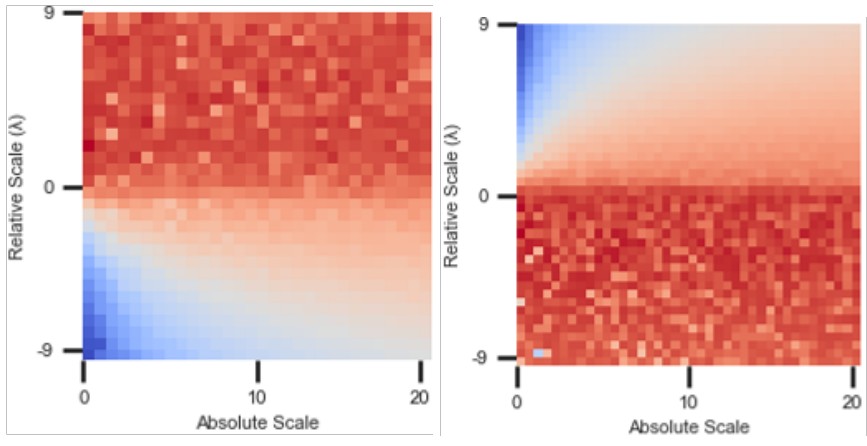

(a) Layer Imbalance in Funnel Network    (b) Layer Imbalance in Anti-Funnel Network

*Figure 10.* **The impact of the architecture.** Linear neural networks that satisfy Equation (14) are trained on randomized regression. Figures show the changes in NTK as a function of weight-to-target ratio and layer imbalances for **(a)**: funnel networks and **(b)**: in anti-funnel networks.

**Grokking in non-linear neural networks**   The study of linear networks does not always directly translate into similar phenomenology in non-linear networks, but these networks successfully highlight the relevant parameters that control

different learning regimes: the relative scale. Notably, networks initialized with upstream scaling can exhibit a delayed entry into the rich learning phase. For example, Kunin et al. (2024) demonstrated that decreasing the variance of token embeddings in a single-layer transformer (accounting for less than 6% of all parameters) significantly reduces the time required to achieve grokking. The method can be viewed as targeted training for the earliest layer that undergoes the most non-linear activations.

### F.4. Additional experimental details for Figure. 6

**Code** The codes used to plot Figure 6 are based on the original codes of Domine et al. (2025) with only minor changes. They are available at `https://github.com/yoonsoonam119/linear_first`.

**Figure. 6(b)** The setup is identical to that of Figure 6(a) except a few changes. We used the linear neural network with $d = 20, p = 20, c = 2$. We randomly initialized $W_2, W_1$ (which do not satisfying the balanced condition) using symmetrized initialization (Chizat et al., 2019) to set the initial function equal to the zero function. The layer imbalance was implemented using the standard deviations $\sigma_1, \sigma_2$ used to initialize the weights. To change the weight-to-target ratio, we used target downscaling in the range $[0.1, 50]$.

# G. Grokking

In this section, we present a brief history and related works on grokking.

Power et al. (2022) first identified grokking, a phenomenon characterized by delayed generalization in algorithmic tasks. Extending beyond algorithmic datasets, Liu et al. (2022b) showed that grokking can also occur in general machine learning tasks, including image classification, sentiment analysis, and molecular property prediction. Humayun et al. (2024) further demonstrated that grokking is a more widespread phenomenon by showing its presence in adversarial robustness for image classification tasks. Wang et al. (2024) discovered that transformers can grok in implicit reasoning tasks, while Zhu et al. (2024) studied the impact of dataset size on grokking in text classification tasks.

## G.1. Approaches to understand grokking

Liu et al. (2022b) claimed that large weight initialization induces grokking. Thilak et al. (2022) proposed the slingshot mechanism to explain grokking, attributing it to anomalies in the adaptive optimization process, particularly in the absence of weight decay. Zheng et al. (2024) argued that sudden generalization is closely related to changes in representational geometry. Levi et al. (2024) addressed grokking in the lazy regime with Gaussian inputs by analyzing gradient-flow training dynamics. Based on this analysis, they demonstrate that grokking time (gap between overfitting ang generalization) depends on weight initialization, input and output dimensions, training sample size, and regularization. However, due to strong assumptions, their analysis is limited to the lazy regime, resulting in an insufficient explanation of other aspects of grokking such as the lazy-to-rich transition.

Several approaches have been proposed to interpret grokking in the modular arithmetic task, where grokking was originally reported. Nanda et al. (2023) and Zhong et al. (2024) used mechanical interpretability to examine case studies of grokking in modular addition. Gromov (2023) provided a solution for grokking in a two-layer MLP on modular addition. He et al. (2024) analyzed grokking in linear modular functions. Mohamadi et al. (2024) demonstrated that a two-layer network can generalize with fewer training data on the modular addition task. Rubin et al.(2024) showed that grokking is a first-order phase transition in both teacher-student setup and modular arithmetic problem.

## G.2. Acceleration of grokking

Several methods have been proposed to accelerate grokking and escape the lazy regime. Liu et al. (2022a) argued that grokking can be mitigated through appropriate hyperparameter tuning, such as adjusting the learning rate or weight decay. Gromov (2023) also suggested that regularization techniques, including weight decay and the momentum of the optimizer, can reduce the training time required for grokking.

Kumar et al. (2024) asserted that **grokking occurs when the network is initialized in the lazy regime, particularly when it has large width, large initialization, small label scaling, or large output scaling.** Lee et al. (2024) proposed amplifying the gradients of low-frequency components to accelerate the speed of generalization. Furuta et al. (2024) suggested pre-training networks on similar tasks that share transferable features, which can then be applied to downstream target tasks to speed up grokking.

## G.3. Empirical confirmation on MLP

**Code**   The code for Figure 8 is available at `https://github.com/yoonsoonam119/linear_first`.

To confirm that our analysis of the scalar input neural network extends to neural networks, we trained depth 4, width 512 MLPs with tanh activation on 1000 datapoints of flattened (vectorized) MNIST dataset. We used MSE loss with the class label as one-hot vector multiplied by the target scale. By default, the weights of the layers were multiplied by factor of 5 compared to LeCun initialization. For training, Adam with learning ratio of $10^{-3}$ and weight decay of $10^{-4}$ was used. Batch size was 128 and the model was trained for 2000 epochs. For input downscaling, the MNIST data vectors were multiplied by the input scaling factor. The output scale was multiplied to the output of the function (MLP). We summarize the different configurations used in Figure 8:

| Name | Sub figure | Weight init ratio | Target scale | Input scale | Output scale |
|---|---|---|---|---|---|
| Default | (a) | 5 | 3 | 1 | 1 |
| Weight downscaling | (b) | **1** | 3 | 1 | 1 |
| Target upscaling | (c) | 5 | **30** | 1 | 1 |
| Input downscaling | (d) | 5 | 3 | **0.01** | 1 |
| Output downscaling | (e) | 5 | 3 | 1 | **0.1** |

# H. Explicit solution for the 2-layer linear neural network with scalar input and scalar output

We provide a general solution for the 2-layer layerwise model described in Equation (15). To repeat the setting, our model and target functions are single variable functions

$$f(x) = \frac{x}{Z} \left( \sum_{i=1}^{p} a_i b_i \right), \quad f^*(x) = xS, \tag{70}$$

where we can set our loss function as

$$\mathcal{L} = \frac{1}{2} \left( \frac{1}{Z} \sum_{i=1}^{p} a_i b_i - S \right)^2. \tag{71}$$

**Theorem 3.** *The solution for the gradient descent dynamics for the loss function (Equation (71)) is given as*

$$a_i(t) = \frac{a_i(0) + b_i(0)}{2} \gamma(t)^{1/2} + \frac{a_i(0) - b_i(0)}{2} \gamma(t)^{-1/2}, \tag{72}$$

$$b_i(t) = \frac{a_i(0) + b_i(0)}{2} \gamma(t)^{1/2} - \frac{a_i(0) - b_i(0)}{2} \gamma(t)^{-1/2}, \tag{73}$$

*where*

$$\gamma(t) = \frac{\Sigma_0 - \mathcal{S}_0 + S + \sqrt{\Sigma_0^2 - \mathcal{S}_0^2 + S^2} + \left( -\Sigma_0 + \mathcal{S}_0 - S + \sqrt{\Sigma_0^2 - \mathcal{S}_0^2 + S^2} \right) \exp\left( -\frac{4\sqrt{\Sigma_0^2 - \mathcal{S}_0^2 + S^2}}{Z} t \right)}{\Sigma_0 + \mathcal{S}_0 - S + \sqrt{\Sigma_0^2 - \mathcal{S}_0^2 + S^2} + \left( -\Sigma_0 - \mathcal{S}_0 + S + \sqrt{\Sigma_0^2 - \mathcal{S}_0^2 + S^2} \right) \exp\left( -\frac{4\sqrt{\Sigma_0^2 - \mathcal{S}_0^2 + S^2}}{Z} t \right)} \tag{74}$$

*and*

$$\mathcal{S}_0 = \frac{1}{Z} \sum_{j=1}^{p} a_j(0) b_j(0), \quad \Sigma_0 = \frac{1}{Z} \sum_{j=1}^{p} \frac{a_j(0)^2 + b_j(0)^2}{2}. \tag{75}$$

*In particular, we have*

$$\gamma_+ := \lim_{t \to \infty} \gamma(t) = \gamma(\infty) = \frac{S + \sqrt{\Sigma_0^2 - \mathcal{S}_0^2 + S^2}}{\Sigma_0 + \mathcal{S}_0}. \tag{76}$$

*The function $\gamma$ is always monotone from $\gamma(0) = 1$ to $\gamma(\infty)$: if $S > \mathcal{S}_0$ then $\gamma$ is increasing, and if $S < \mathcal{S}_0$ then $\gamma$ is decreasing.*

*Proof.* Since our gradient descent dynamics is given as

$$\frac{da_i}{dt} = -\frac{\partial \mathcal{L}}{\partial a_i} = -b_i \left( Z^{-2} \sum_{j=1}^{m} a_j b_j - Z^{-1} S \right), \quad \frac{db_i}{dt} = -\frac{\partial \mathcal{L}}{\partial b_i} = -a_i \left( Z^{-2} \sum_{j=1}^{m} a_j b_j - Z^{-1} S \right), \tag{77}$$

one can observe

$$\frac{d}{dt}(a_i + b_i) = -(a_i + b_i) \left( Z^{-2} \sum_{j=1}^{p} a_j b_j - Z^{-1} S \right), \quad \frac{d}{dt}(a_i - b_i) = (a_i - b_i) \left( Z^{-2} \sum_{j=1}^{p} a_j b_j - Z^{-1} S \right). \tag{78}$$

So if we define $\gamma$ as

$$\gamma(t) = \exp\left( -2 \int_{s=0}^{t} \left( Z^{-2} \sum_{j=1}^{p} a_j(s) b_j(s) - Z^{-1} S \right) ds \right), \tag{79}$$

then we have

$$\frac{d}{dt}(\gamma) = -2\gamma \left( Z^{-2} \sum_{j=1}^{p} a_j b_j - Z^{-1} S \right). \tag{80}$$

Hence

$$\frac{d}{dt}\left(\gamma^{-1/2}(a_i + b_i)\right) = \gamma^{-1/2}\left(Z^{-2}\sum_{j=1}^{p}a_jb_j - Z^{-1}S\right)(a_i + b_i) - \gamma^{1/2}\left(Z^{-2}\sum_{j=1}^{p}a_jb_j - Z^{-1}S\right)(a_i + b_i) = 0,$$

(81)

$$\frac{d}{dt}\left(\gamma^{1/2}(a_i - b_i)\right) = -\gamma^{1/2}\left(Z^{-2}\sum_{j=1}^{p}a_jb_j - Z^{-1}S\right)(a_i - b_i) + \gamma^{1/2}\left(Z^{-2}\sum_{j=1}^{p}a_jb_j - Z^{-1}S\right)(a_i - b_i) = 0.$$

(82)

From those and $\gamma(0) = 0$ we have

$$a_i(t) + b_i(t) = (\gamma(t))^{1/2}(a_i(0) + b_i(0)), \quad a_i(t) - b_i(t) = (\gamma(t))^{-1/2}(a_i(0) - b_i(0)),$$

(83)

which leads to Equation (72) and Equation (73). We next work on dynamics of $\gamma$. We start from

$$\frac{1}{\gamma}\frac{d\gamma}{dt} = \frac{d}{dt}(\log\gamma) = -2Z^{-2}\sum_{j=1}^{m}a_jb_j + 2Z^{-1}S.$$

(84)

We can express

$$Z^{-2}\sum_{j=1}^{p}a_jb_j = Z^{-2}\sum_{j=1}^{p}\left(\frac{(a_j + b_j)^2}{4} - \frac{(a_j - b_j)^2}{4}\right)$$

(85)

$$= Z^{-2}\sum_{j=1}^{p}\frac{(a_j(0) + b_j(0))^2}{4}\gamma - Z^{-2}\sum_{j=1}^{p}\frac{(a_j(0) + b_j(0))^2}{4}\gamma^{-1}$$

(86)

$$= Z^{-2}\sum_{j=1}^{p}a_j(0)b_j(0)\frac{\gamma + \gamma^{-1}}{2} + Z^{-2}\sum_{j=1}^{p}\frac{a_j(0)^2 + b_j(0)^2}{2}\frac{\gamma - \gamma^{-1}}{2}.$$

(87)

So if we denote

$$\mathcal{S}_0 = \frac{1}{Z}\sum_{j=1}^{p}a_j(0)b_j(0), \quad \Sigma_0 = \frac{1}{Z}\sum_{j=1}^{p}\frac{a_j(0)^2 + b_j(0)^2}{2},$$

(88)

then our dynamics for $\gamma$ becomes

$$\frac{d\gamma}{dt} = -2Z^{-1}\gamma\left(\mathcal{S}_0\frac{\gamma - \gamma^{-1}}{2} + \Sigma_0\frac{\gamma + \gamma^{-1}}{2} - S\right) = -2Z^{-1}\left((\Sigma_0 + \mathcal{S}_0)\gamma^2 - 2S\gamma - (\Sigma_0 - \mathcal{S}_0)\right).$$

(89)

One can observe

$$\Sigma_0 = \frac{1}{Z}\sum_{j=1}^{p}\frac{a_j(0)^2 + b_j(0)^2}{2} \geq \frac{1}{Z}\sum_{j=1}^{p}|a_j(0)b_j(0)| \geq |\mathcal{S}_0|,$$

(90)

so we can always factor the quadratic in $\gamma$ as

$$(\Sigma_0 + \mathcal{S}_0)\gamma^2 - 2S\gamma - (\Sigma_0 - \mathcal{S}_0) = (\Sigma_0 + \mathcal{S}_0)(\gamma - \gamma_-)(\gamma - \gamma_+)$$

(91)

where

$$(\gamma_-, \gamma_+) = \left(\frac{S - \sqrt{\Sigma_0^2 - \mathcal{S}_0^2 + S^2}}{\Sigma_0 + \mathcal{S}_0}, \frac{S + \sqrt{\Sigma_0^2 - \mathcal{S}_0^2 + S^2}}{\Sigma_0 + \mathcal{S}_0}\right) \quad \Rightarrow \quad \gamma_- < 0 < \gamma_+.$$

(92)

We continue using partial fraction decomposition as

$$\frac{1}{(\gamma - \gamma_-)(\gamma - \gamma_+)}\frac{d\gamma}{dt} = \frac{1}{\gamma_+ - \gamma_-}\left(\frac{1}{\gamma - \gamma_+} - \frac{1}{\gamma - \gamma_-}\right)\frac{d\gamma}{dt} = -2Z^{-1}(\Sigma_0 + \mathcal{S}_0)$$

(93)

and

$$\frac{d}{dt} \log \left| \frac{\gamma - \gamma_+}{\gamma - \gamma_-} \right| = -2Z^{-1}(\Sigma_0 + \mathcal{S}_0)(\gamma_+ - \gamma_-) = -4Z^{-1}\sqrt{\Sigma_0^2 - \mathcal{S}_0^2 + S^2}. \tag{94}$$

Both $\gamma - \gamma_+$ and $\gamma - \gamma_-$ do not change signs, so the following holds regardless of the sign of $\gamma - \gamma_+$:

$$\frac{\gamma - \gamma_+}{\gamma - \gamma_-} = \frac{\gamma(0) - \gamma_+}{\gamma(0) - \gamma_-} \exp\left( -\frac{4\sqrt{\Sigma_0^2 - \mathcal{S}_0^2 + S^2}}{Z} t \right). \tag{95}$$

As $\gamma(0) = 1$, this is solved as

$$\gamma(t) = \frac{\Sigma_0 - \mathcal{S}_0 + S + \sqrt{\Sigma_0^2 - \mathcal{S}_0^2 + S^2} + \left(-\Sigma_0 + \mathcal{S}_0 - S + \sqrt{\Sigma_0^2 - \mathcal{S}_0^2 + S^2}\right) \exp\left(-\frac{4\sqrt{\Sigma_0^2 - \mathcal{S}_0^2 + S^2}}{Z} t\right)}{\Sigma_0 + \mathcal{S}_0 - S + \sqrt{\Sigma_0^2 - \mathcal{S}_0^2 + S^2} + \left(-\Sigma_0 - \mathcal{S}_0 + S + \sqrt{\Sigma_0^2 - \mathcal{S}_0^2 + S^2}\right) \exp\left(-\frac{4\sqrt{\Sigma_0^2 - \mathcal{S}_0^2 + S^2}}{Z} t\right)}. \tag{96}$$

The limit of $t \to \infty$ in Equation (95) gives

$$\gamma(t) \xrightarrow{t \to \infty} \gamma_+ = \frac{S + \sqrt{\Sigma_0^2 - \mathcal{S}_0^2 + S^2}}{\Sigma_0 + \mathcal{S}_0}. \tag{97}$$

From Equation (95), we can also see $\gamma \in (\gamma_-, \gamma_+)$, and $\gamma$ increases if and only if $1 - \gamma_+ < 0$. It is straightforward to check that if $S > \mathcal{S}_0$ (or $S < \mathcal{S}_0$) then $\gamma_+ > 1$ (and $\gamma_+ < 1$ respectively). $\qquad \square$

*Remark* 4. If $S > \mathcal{S}_0$, then the graph of $\gamma(t)$ becomes part of some sigmoid function whose image is $(\gamma_-, \gamma_+)$ and center is at height $\frac{\gamma_- + \gamma_+}{2} = \frac{S}{\Delta_0 + \Sigma_0}$. As the graph of $\gamma$ starts from height $1 = \gamma(0)$, one can show that $\gamma(t)$ is concave for all $t \geq 0$ if and only if $S < \Delta_0 + \Sigma_0$, and otherwise it contains an inflection point. If $S < \mathcal{S}_0$, the graph takes a shape of (dilation and translation of) $(1 - e^{-t})^{-1}$, so after the point $\gamma(0) = 1$ it is convex and monotone decreasing.

*Remark* 5. Since $a_i(0)^2 + b_i(0)^2 \geq 2a_i(0)b_i(0)$, we have $\Sigma_0 \geq 0$ and $\Sigma_0 \geq \mathcal{S}_0$. Therefore,

$$\gamma_+ = \gamma(\infty) = \frac{S + \sqrt{\Sigma_0^2 - \mathcal{S}_0^2 + S^2}}{\Sigma_0 + \mathcal{S}_0} \geq \frac{2S}{2\Sigma_0} = \frac{S}{\Sigma_0} = \left(\frac{\Sigma_0}{S}\right)^{-1},$$

Hence, if $\Sigma_0/S$ is small, we have large $\gamma_+$ (rich regime).

# I. Research directions

We share additional research directions for layerwise linear models.

## I.1. Connection between generalization and greedy dynamics

Greedy dynamics or the rich regime often correlate with generalization (Lampinen & Ganguli, 2019; Xu & Ziyin, 2024). Generalizing models in vision tasks exhibit NC (Papyan et al., 2020). Models that grok achieves a generalizing solution when they are in the rich regime (Kumar et al., 2024).

However, greedy dynamics alone cannot fully predict generalization. ResNet18 can show NC on CIFAR10 dataset with partially and fully randomized labels (Zhu et al., 2021). Transition into a rich regime in grokking may degrade the performance (Lyu et al., 2024). The imperfect correlation between greedy dynamics and generalization highlights the need for a deeper understanding of what these dynamics optimize.

## I.2. Encouraging greedy dynamics

One of the goals of understanding grokking is to uncover techniques for transitioning models into generalizing ones. In this paper and previous studies, we have gained insight into the rich regime and ways to enhance it. In fact, extreme rich regimes have shown improved performance on certain tasks (Atanasov et al., 2025). Can this knowledge be extended to promote rich regimes in various scenarios, such as during training?

