# OpenReview forum: "Position: Solve Layerwise Linear Models First to Understand Neural Dynamical Phenomena (Neural Collapse, Emergence, Lazy/Rich Regime, and Grokking)"
_ICML.cc/2025/Position_Paper_Track — ICML 2025 Position Paper Track poster_

### Official Review · Reviewer_7z5u · 2025-03-08

**Significance:** 3
**Argument Clarity:** 2
**Rating:** 2
**Confidence:** 3

**Questions:**

Can you give a definition for each of the phenomena? A mathematical definition (at least for the linear models) would significantly improve the presentation.

Are the connections between the different phenomena mathematically precise or conjectural? Can these be written in the form of a theorem?

**Discussion Potential:**

2

**Paper Summary:**

The paper advocates for studying the dynamics of gradient flow in machine learning models by analyzing its behavior in linear layers. The paper claims that different phenomena like neural collapse, grokking, and emergence can be studied in the same mathematical framework. The paper has a short section at the end giving a high-level description of two research directions, and other directions are summarized in an appendix.

**Position:**

Yes

**Position In Title:**

Yes

**Related Work:**

3

**Strengths And Weaknesses:**

Studying the behavior of complicated models by instead analyzing a simplified version is a common mathematical approach. As the authors mention, a rich literature studies different phenomena empirically identified in deep learning in a linear setting.
This paper connects all these phenomena in the linear case via the dynamics feedback principle, which is a set of properties of gradient flow for models defined as products of linear layers.

In my opinion, this paper is not a good fit for the position paper track. Conceptually it would work better as a research paper or a survey paper.

Moreover, the paper does not do a good job of explaining how all these different phenomena can be seen as a consequence of the dynamic feedback principle. It is unclear if the connection is established in the cited papers or conjectural (and therefore part of the research program). At the minimum, the main document should give precise definitions of the phenomena it tries to explain. If it is not possible to give a full definition, it at least should define them in the linear setting.

**Support:**

2

---

> ### Author Rebuttal · Authors · 2025-03-28
>
> Thank you for your careful review.
>
> We appreciate your perspective on whether this work is best suited as a survey or research paper— a key point in our internal discussions. While both were viable options, we chose a position paper to address the underappreciation of layerwise linear models.
>
>  ### **Why Position**
> Our main **contribution** is the **perspective** on these models (i.e., the dynamical feedback principle), with the **goal** of **convincing** the community of their usefulness. A research paper, emphasizing technical contributions, would shift focus away from this perspective, while a survey, requiring neutrality, would prevent us from taking a strong stance. A position paper, therefore, was the ideal format.
>
> A complete theory consists of an intuitive guiding principle, formal analysis, and empirical support. Research papers typically address the latter two, so we **focused on the first: providing an intuitive justification**. While rigor enhances clarity, it can also hinder accessibility. We prioritized intuition to engage a **broader audience with minimal mathematical barriers**.
>
> We adopted a top-down approach rather than a definition-lemma-theorem structure. Each section (4-7) begins by linking back to the dynamical feedback principle and the toy model in Figure 1, introduces a solvable minimal setup, connects it to related work, and then compares it to a practical phenomenon. Formal details are deferred to the appendices.
>
>  ### **Definitions**
> To ensure clarity despite sacrificing some rigor in the main text, we have:
>  * **Glossary in Appendix A**: Definitions of key terms and phenomena, alongside references to where they appear in the main text and appendices (e.g., neural collapse is formally defined in Appendix E). However, **many phenomena remain open research questions without agreed-upon definitions (even in linear case)**. For instance, Anwar et al. 2024 (https://arxiv.org/abs/2404.09932, Section 2.3.4) notes that "emergence" lacks a precise definition.
>  * **Intuitive descriptions with references**: We provided our most eloquent descriptions (e.g., grokking as a delayed generalization) with **reference** to key studies, though definitions vary in level of formality.
>  * **Literature reviews in Appendices D-G**: We have dedicated appendices covering the current understanding of each phenomenon discussed in Sections 4-7.
>
>  ### **Planned Revisions**
> Your feedback closely aligns with our writing considerations, and we appreciate your valuable insights. While our work balances accessibility and rigor, we aim to refine our explanations further by:
>  * Expanding Section 1.2 to emphasize glossary.
>  * Extracting and enhancing definitions out from the glossary in Appendix A into a separate dictionary.
>  * Explicitly referencing cited paper's theorem or section numbers
>  * Formalizing derivations into propositions (e.g., derivations in C.1 and C.2)
>
>  ## Question Response
> Phenomena often lack strict mathematical definitions, but we reference key papers where they are formally introduced and where we introduce them in our paper:
>  * Emergence – Wei et al. 2022 (https://arxiv.org/abs/2206.07682) [Appendix D in our paper]
>  * Neural Collapse – Papyan et al. 2020 (https://arxiv.org/abs/2008.08186) [Appendix E  in our paper]
>  * Lazy/Rich Regime – Chizat et al. 2019 (https://arxiv.org/abs/1812.07956) [Section 6  in our paper]
>  * Grokking – Power et al. 2022 (https://arxiv.org/abs/2201.02177) [Appendix G  in our paper]
>
> To the best of our knowledge, no single theorem connects these results, as they rely on different assumptions and setups. A universal theorem covering the complete dynamics of all two-layer networks with general initialization, along with concrete definitions for various phenomena, remains an **ongoing and evolving discussion**. At this stage, such a result is far-fetched. Instead, the cited works focus on specific, solvable cases. We believe the dynamical feedback **principle offers a broad conceptual guideline**, while **theorems rigorously establish its implications**—hence the green and blue levels in Figure 1.
>
> The dynamical feedback principle groups recurring similarities in theorems found in the citations. For example:
> Saxe et al. 2013 (https://arxiv.org/abs/1312.6120) studied the non-linear dynamics of linear networks.
> Mixon et al. 2020 (https://arxiv.org/abs/2011.11619) studied neural collapse in an unconstrained feature model.
> Nam et al. 2024 (https://arxiv.org/abs/2404.17563) studied emergence in a multilinear model.
>
> At first glance, these results describe different models, making them seem incompatible. However, Theorem 3 in Mixon et al., Lemma 1 in Nam et al., and equations 8-12 in Saxe et al. show that their dynamics are isomorphic as dynamical systems (apart from dimensional differences) to the sigmoidal dynamics ("hyperbolic dynamics" in Saxe et al.). Opening such perspective was our major contribution.

---

### Official Review · Reviewer_iFRq · 2025-03-13

**Significance:** 4
**Argument Clarity:** 4
**Rating:** 5
**Confidence:** 3

**Questions:**

For figure four I could only understand from your time dynamics solution that the skills are learned at different times because of the sigmoid loss. However you haven't explained why data emergence and parameter emergence happy? Namely why once we increase time or increase data, the skill strength could be improved significantly? Is there an explanation connected with your dynamic of feedback principle?

**Discussion Potential:**

4

**Paper Summary:**

This paper introduces the Dynamical Feedback Principle, a novel framework for understanding and analyzing the behavior of neural networks. The authors develop this principle by systematically studying diagonal linear neural networks and fully linear neural networks. By deriving time-dependent differential equations from first principles, they characterize the trajectory of parameters in these models.

The Dynamical Feedback Principle serves as a high-level guiding framework that explains various observed phenomena in neural networks, including the sigmoidal growth of neurons, neuron collapse, and the impact of imbalanced parameter initialization on learning dynamics. Additionally, it provides insights into the different growth phases of neural networks. The authors further demonstrate that careful initialization of parameters can significantly influence learning rates and overall model accuracy. This work offers a foundational perspective on neural network training dynamics and proposes a theoretical basis for improving optimization strategies.

## update after rebuttal

I have read all reviews and rebuttal comments for this paper. Overall, I think this is a very strong position paper with good, clear writing as well as solid mathematical elements. The topic is also very relevant for today's research --- understanding the phenomenon of deep learning by studying some simplified models first and trying to extract some insights from these simple models.

The reviews giving low scores either are a little bit too generic or do not point out any specific crucial weakness for reason of rejection.

I therefore would like to maintain my score and champion for the acceptance of this paper.

**Position:**

Yes

**Position In Title:**

Yes

**Related Work:**

4

**Strengths And Weaknesses:**

This paper is well-written and presents its concepts clearly. The Dynamical Feedback Principle is well-articulated, and the authors effectively illustrate different phenomena using detailed figures, which aid in conveying high-level ideas. Despite not specializing in theoretical deep learning, I found the paper accessible, suggesting that it can serve as a useful introduction to these topics for a broader audience. Additionally, the paper has the potential to be used as a reference for future research on neural network dynamics.

One key limitation of the work is that the layer-wise neural network models studied do not include nonlinear activation functions, creating a gap between the theoretical framework and practical deep learning models. The absence of nonlinearity raises concerns about the applicability of the proposed principles to real-world networks, where activation functions play a crucial role in expressivity and learning dynamics. However, given that significant theoretical progress in analyzing deep networks without activation functions is still lacking, this study on simplified models remains valuable. It lays the groundwork for future research that could extend these principles to more realistic architectures.

Overall, while the study provides valuable theoretical insights, bridging the gap between these toy models and practical neural networks remains an open challenge for future work.

**Support:**

4

---

> ### Author Rebuttal · Authors · 2025-03-30
>
> Thank you for your thoughtful and generous review.
>
> We are delighted that you found the paper accessible, as making it more approachable was one of our key goals. We agree that layerwise linear models are not a complete description of practical neural networks; they serve as a plausible starting point—capturing various phenomena despite their simplicity.
> We believe this framework will be a stepping stone for more complex theories, helping future researchers distinguish whether a phenomenon arises from less understood non-linearity or the more structured dynamics of layerwise models.
> Once again, we sincerely appreciate your time and effort in reviewing our work.
>
>  ## Question Response:
>
> This is a great question! We discuss this in Appendix D.4, following the argument in Nam et al. 2024 (https://arxiv.org/abs/2404.17563). However, since the reasoning is less straightforward than time emergence, we deferred it to the appendix and reference. We will improve Appendix D.4 to make the paper more self-contained.
>
> Re-iterating the argument in Nam et al., two key steps are required for understanding the connection between the dynamical feedback principle and data/parameter emergence:
>  * **Justifying the multilinear model as a valid approximation of practical DNNs.**
>  * **Solving the multilinear model’s dynamics for data and parameter emergence.**
>
>
>  ### Step 1: Layerwise Dynamics and the Multilinear Model
> While **DNNs do not inherently separate skills**, multilinear models (Eq. 11 in our paper) are aware of different skills due to the prebuilt skill functions. As we will later argue in step 2, the multilinear models shows data and parameter emergence due to the skill functions and the decoupled dynamics that stem from it. The question is now **how DNNs acknowledge/separate the skills**. Authors of Nam et al. justify the behavior through the **layerwise dynamics** (dynamical feedback in our paper). The layerwise structure and the power-law in skill frequency lead to **stage-like training** where skills emerge in distinct phases and the dynamics of learning the **skills remain effectively decoupled** (e.g., Figure 4(a)).
>
> Intuitively, DNNs **prioritize (feature-)learning frequent skills under resource constraints**. In parameter emergence, limited (hidden) neurons are allocated to express frequent skills only, creating a sharp performance gap between more and less frequent skills. Data emergence follows similar pattern: more training examples of a skill ($d_k$ in Nam et al.) accelerate feature-learning of the skill, allowing frequent skills to become distinguishable (better skill performance) while infrequent ones remain entangled (poorer skill performance). Given that skill frequencies follow a power-law, a significantly more training set is required to achieve a critical number of samples for scarce skills. Thus, scarce skills emerge only at significantly larger training set size. Also, see the limitations section in Nam et al. for limitations of these approaches.
>
>  ### Step 2: Data and Parameter Emergence in the Multilinear Model
> Here, we will brief how prebuilt skill functions lead to emergence. Please see appendices of Nam et al. for a detailed analysis, especially Corollary 1 and 2 and Propositions 1 and 2.
>  ### Data Emergence
> Corollary 1 of Nam et al. states that a multilinear model is a one-shot learner, meaning a single observation of the $k^{th}$ skill, for infinite training time and infinite parameters, leads to a training of the $k^{th}$ skill: \
> $a_k(\infty)b_k(\infty) = S$ if $d_k >0,$  $0$ elif $d_k = 0,$\
> where $d_k$ is the number of training examples of the $k^{th}$ skill. Thus, the probability that a sample from the $k^{th}$ skill exists in the training set determines the expected saturation $E[a_k(\infty)b_k(\infty)]$. Because of the power-law in skill frequency, this value shows a relatively abrupt change of training set size in log-scale, and thus data emergence. Authors of Nam et al. extended this model to a $D_c$ shot-learner instead of one-shot learner in Proposition 1 and used the model to fit Figure 4(b) in our paper.
>
>  ### Parameter Emergence
> Corollary 2 states that because the skill functions are mutually exclusive, the model will learn the $k^{th}$ skill - for an infinite training set and time - whenever the $k^{th}$ skill function $g_k$ exists in the model. Formally, \
> $a_k(\infty)b_k(\infty) = S$ if $k \leq N,$ $0$ elif $k > N,$ \
> where $N$ is the width of the diagonal neural network ($p$ in our paper). The parameter emergence is trivial as the model will suddenly learn a skill once corresponding skill function is added. Nam et al. extended this model, requiring $N_c$ hidden neurons to express a skill, and fit it to Figure 4(c) in our paper.

---

> > ### Comment · Reviewer_iFRq · 2025-04-04
> >
> > Thank you very much for the answers.
> >
> > I would recommend putting some of these explanations in the main paper, or at least writing a sentence pointing to D.4. in the Appendix.

---

> > > ### Author Response · Authors · 2025-04-05
> > >
> > > Thank you for your comments. We wholeheartedly agree with your suggestions and will revise accordingly:
> > >  * In Subsection 4.3, describe how skill functions lead to data and parameter emergence in the multilinear model and add the reference to Appendix D.4.
> > >  * In the intuition paragraph of Section 4, expand our description on effective decoupling of the skills to explicitly include both data and parameter emergence.
> > >  * In Appendix D.4, expand the whole description for better clarity (as mentioned in the rebuttal).

---

### Official Review · Reviewer_qWr3 · 2025-03-14

**Significance:** 2
**Argument Clarity:** 2
**Rating:** 2
**Confidence:** 4

**Questions:**

New DNN architectures and components have been developed to solve new challenges in practical problems, such as the architectures using graph or distributed structures. These structures have been utilized together with layerwise models but not necessarily dependent on them. Then, how does your proposal explicate dynamics of these DNNs as well?

**Discussion Potential:**

2

**Paper Summary:**

This position paper suggests using layerwise linear models retaining the core principles of neural dynamical phenomena to accelerate the science of deep learning. More precisely, the paper argues that new studies should prioritize the dynamics of layerwise structure as the key feature over the non-linear activations, until its limits are fully explored.

To this end, the paper introduces a systematic presentation of various works on layerwise linear models with a dynamical feedback mechanism. Then, they discuss how this property from the layerwise linear model extends to describe an empirical phenomenon in DNNs.

## Update after rebuttal

I checked all responses and comments from authors and reviewers. Some of my questions were addressed in the rebuttal. However, I consider that some of the claims in the paper should be revised as discussed in the rebuttal by the authors (e.g. generalization of the proposed theory to cover a large family of learning methods and models). Therefore, I keep my score.

**Position:**

Yes

**Position In Title:**

Yes

**Related Work:**

1

**Strengths And Weaknesses:**

The paper states a nice discussion on utilizing a dynamical feedback mechanism to explore various characteristics of layerwise linear models.

The proposed systematic presentation of various works on layerwise linear models is interesting but limited, since it does cover the works on utilizing linear models for other learning regimes such as semi-supervised, self-supervised, and un-supervised learning with different data regimes.

Indeed, understanding dynamics of layerwise linear models is crucial for elucidating the underlying mechanisms of DNNs. However, DNNs have been evolving with new architectures in new use-cases and applications. Therefore, the main proposal on prioritizing “the dynamics of layerwise structure as the key feature over the non-linear activations, until its limits are fully explored” is a confined and stoic approach.

**Support:**

2

---

> ### Author Rebuttal · Authors · 2025-03-28
>
> Thank you for your review.
>
> While our theoretical models focus on supervised learning with layerwise linear structures, the models **qualitatively explain the phenomena that extend beyond linear neural networks in supervised learning setup**. For example, emergence (https://arxiv.org/abs/2206.07682) and scaling laws (https://arxiv.org/abs/2001.08361) appear in LLMs trained with both supervised and unsupervised learning. Neural collapse (https://arxiv.org/abs/2008.08186) is observed in CNNs for vision classification, and grokking (https://arxiv.org/abs/2201.02177) emerges in transformers trained on arithmetic tasks. Having said that, we acknowledge the need to emphasize the significance a bit more, in which we thank the reviewer.
>
> We believe the ability to qualitatively explain such diverse phenomena across architectures and datasets **without intricate details** is **a strength, not a limitation** of the principle. We argue that **a key characteristic shared by all DNNs**—the dynamics of gradient descent under layerwise structure—enables these simple models to make broad qualitative predictions. Our position is that understanding this fundamental principle should take **priority** before focusing on intricate details such as architectures, as they will guide theoreticians and engineers in determining whether a problem (e.g., delay in generalization) arises because of bad initialization (lack of amplifying dynamics) or less trivial factor specific to architecture or dataset.
>
> At the same time, as we discussed in the alternative view section, we respect your perspective. Practical neural networks rely on nonlinearities, engineering tricks, architecture, and task-specific tuning. However, no single theory can fully capture all aspects of real-world models—it would be too complex or intractable. We believe that theoretical progress often starts with core principles, and our aim is to highlight how simpler models capture this principle effectively.
>
> Additionally, we do not claim that layerwise dynamics alone form a complete theory of deep learning. Rather, we see it as a **necessary step before developing more complex theories**. For example, we agree that architecture is undoubtedly important in practice. We referenced relevant works, such as Pinson et al. 2023 for CNNs (https://arxiv.org/abs/2303.02034) and Ahn et al. 2023 for transformers (https://arxiv.org/abs/2310.01082), and discuss that these are valid directions for future work.
>
> In summary, while factors like architecture, normalization, and data structure are crucial, we advocate for first understanding simpler, more generalizable models before tackling complex refinements. A strong theoretical foundation prevents unnecessary overcomplication and provides insights that benefit both theoreticians and practitioners. Moving on, we will update our paper to emphasize these points both as limitations and avenues for future work.
>
>  ## Question Response:
>
> To the best of our knowledge, **only a small fraction** of practical topics, such as prompt engineering, are **independent of layerwise structure and gradient descent**, which is also the scope of our paper. This shared foundation makes the dynamical feedback principle—and the layerwise linear models that capture it—highly relevant across diverse architecture-task setups.
>
> Again, we do not claim it as a complete theory of all DNNs, explaining all advances in the community. Rather, we see it as a plausible and often solvable framework that should be considered before resorting to more complex alternatives, helping both theoreticians and practitioners avoid unnecessary overcomplication.

---

### Official Review · Reviewer_rVo6 · 2025-03-14

**Significance:** 4
**Argument Clarity:** 4
**Rating:** 5
**Confidence:** 3

**Questions:**

How would the optimisation method affect the results? Do the solutions/evolution of parameters remain the same when one uses RMSProp, Adam, or second-order methods? If not would it be possible to show a simple example of it in an appendix for future study?

**Discussion Potential:**

3

**Paper Summary:**

The paper argues for the study of Layerwise Linear Models as a way to understand Deep Learning Phenomena, by showing that they exhibit interesting non-linear dynamics during training, which are in many cases analytically solvable, and that by selecting some specific properties one can use them to study many of the important or counter-intuitive properties of Deep Neural Networks, like Scaling Laws, Grokking, Emergence and others.

In the end there's a call to action for analysing the importance of depth in layerwise linear systems, and for looking into how to model non-linearities by using pertubation methods, a common technique in physics.

## update after rebuttal
I still believe this is a strong paper and will keep my high rating

**Position:**

Yes

**Position In Title:**

Yes

**Related Work:**

4

**Strengths And Weaknesses:**

The paper is well-situated on the literature of Deep Learning Phenomena and of Deep Linear models, it connects itself with some of the main interesting or counter-intutive properties of Deep Neural Networks, and it has rigorous but easy to understand mathematical arguments for the importance of layerwise linear networks.

One of the main weakness of the paper is that it does not deal with depth, which limits how effective their method can be at analysing things like the Lottery Ticket Hypothesis, and also that things like the effect of the chosen optimization method (SGD vs Adam vs Second Order etc) are not mentioned.

**Support:**

3

---

> ### Author Rebuttal · Authors · 2025-03-28
>
> We thank you for your detailed and constructive review.
>
> Your summary well captures our paper and our motivation. We especially thank you for mentioning depth as one of the limitations of the current framework. We mainly focused on two-layer systems for minimal setup, but we agree a linear neural network with depth can aid the readers’ understanding, and indeed make a plausible connection to lottery ticket hypothesis and deep neural collapse.
>
> Depth, by having more product of parameters, transforms the sigmoidal dynamics into more step-like function, at least for diagonal neural networks. The product of parameters results in more abrupt saturation and strengthens the stage-like training. Related works on depth include Saxe et al. 2013 (https://arxiv.org/abs/1312.6120), more recently Lyu et al. 2023 (https://dl.acm.org/doi/10.5555/3666122.3668658) and Kunin et al. 2024 (https://arxiv.org/abs/2406.06158). We acknowledge that depth stands as an important direction for future work, and aim to introduce a discussion on the topic once the space limitations are more relaxed.
>
> Although optimization is an interesting topic on its own, we did not explore the connection fully in this paper and focused on gradient flow without noise. We focused on higher level phenomena where the role of optimizers and second order methods are less critical. For example, we trained grokking with Adam in appendix G, but the results qualitatively agree with analysis of Section 7 and Appendix H which assumes gradient flow.
>
> Overall, we thank you again for a constructive review and appreciating our work on how layerwise linear models explain various phenomena of practical DNNs. We will include depth into the current work and explore the connection to optimizers for future work.
>
>
>
>  ## Question Response:
> The role of optimization methods is an interesting field, but we unfortunately have not had the chance to make an intuitive connection with dynamical feedback principle. We found – not stated in the paper – that Adam can sometimes make the dynamics more rich (tighter neural collapse) in ResNets and VGG16s but more lazy for MLPs trained on CIFAR10. Due to this observation, and lack of clear connection through a solvable model, we have not explored the topic further and left this for future work.
>
> Although this requires proper analysis, we share our best speculation. So please take this with a large grain of salt. We speculate that second order methods, by making the landscape less flat in certain directions, promote feature learning by making more features accessible. This may explain our observation on ResNet18 and MLPs. With access to more features, ResNet18 may benefit from better feature-learning and satisfy the free unconstrained assumption in unconstrained feature model: that the last features can express the training set without any constraints on dynamics. MLP, with less suitable architecture for feature learning, may benefit less and end up simply using more features (i.e. deviate from neural collapse). However, this is still only a speculation left for further study.

---

### Decision · Program_Chairs · 2025-04-30

**Decision:**

Accept (poster)

**Comment:**

Two of the reviewers recommend strong acceptance and two recommend weak rejection. After reading all the reviews, the authors's rebuttal, and the paper, I agree with the arguments presented by the reviewer who decided to champion the acceptance of the paper. This is a well-written strong position paper, with solid mathematical arguments, on a topic that is very relevant today: understanding deep learning by studying some simplified models first and trying to extract some insights from these simple models, in the well-proven tradition of the spherical cow approach (https://en.wikipedia.org/wiki/Spherical_cow).